# Subthalamic stimulation modulates context-dependent effects of beta bursts during fine motor control

Manuel Bange [1], Gabriel Gonzalez-Escamilla [1], Damian M. Herz [1,2], Gerd Tinkhauser [3], Martin Glaser[4], Dumitru Ciolac [1], Alek Pogosyan[2], Svenja L. Kreis [5], Heiko J. Luhmann [5], Huiling Tan[2] & Sergiu Groppa [1] ✉

Increasing evidence suggests a considerable role of pre-movement beta bursts for motor control and its impairment in Parkinson's disease. However, whether beta bursts occur during precise and prolonged movements and if they affect fine motor control remains unclear. To investigate the role of within-movement beta bursts for fine motor control, we here combine invasive electrophysiological recordings and clinical deep brain stimulation in the subthalamic nucleus in 19 patients with Parkinson's disease performing a context-varying task that comprised template-guided and free spiral drawing. We determined beta bursts in narrow frequency bands around patient-specific peaks and assessed burst amplitude, duration, and their immediate impact on drawing speed. We reveal that beta bursts occur during the execution of drawing movements with reduced duration and amplitude in comparison to rest. Exclusively when drawing freely, they parallel reductions in acceleration. Deep brain stimulation increases the acceleration around beta bursts in addition to a general increase in drawing velocity and improvements of clinical function. These results provide evidence for a diverse and task-specific role of subthalamic beta bursts for fine motor control in Parkinson's disease; suggesting that pathological beta bursts act in a context dependent manner, which can be targeted by clinical deep brain stimulation.

The execution of complex movements like walking or drawing is commonly impaired in patients with Parkinson's disease (PD), posing a substantial burden on the quality of life of the affected person[1–3]. Because the mechanisms underlying such impairments might be multidimensional and remain unclear, better characterization is needed to optimize current clinical interventions. A well-established and highly effective therapy for managing the cardinal symptoms like bradykinesia and tremor consists of delivering high-frequency electrical stimulation to subcortical neural structures (deep brain stimulation, DBS) such as the subthalamic nucleus (STN)[4–6]. Although widespread adoption of DBS motivated efforts to further improve its effectiveness, efficiency, and therapeutic window, the effects of DBS on the execution of complex every-day movements are poorly understood. In contrast to the highly standardized motor assessment in clinical routine, complex movements better represent tasks a patient might encounter during their usual day. Such daily activities

[1]Section of Movement Disorders and Neurostimulation, Department of Neurology, Focus Program Translational Neuroscience (FTN), University Medical Center of the Johannes Gutenberg-University Mainz, Mainz, Germany. [2]MRC Brain Network Dynamics Unit, Nuffield Department of Clinical Neurosciences, University of Oxford, Oxford, UK. [3]Department of Neurology, Bern University Hospital and University of Bern, Bern, Switzerland. [4]Department of Neurosurgery, University Medical Center of the Johannes Gutenberg-University Mainz, Mainz, Germany. [5]Institute of Physiology, University Medical Center of the Johannes Gutenberg-University Mainz, Mainz, Germany. ✉e-mail: segroppa@uni-mainz.de

oftentimes require the skilled and fine coordination of multiple selected effectors, as well as the continuous integration of feedback and dynamic adjustment of movements[7,8]. Drawing, for example, is a graphomotor skill involving the precise control of finger-, hand-, and arm-muscles, as well as the continuous integration of visual and kinesthetic feedback[9,10]. In PD, drawing movements display a variety of abnormalities including reduced drawing velocity, increased velocity fluctuations, an elevated number of velocity and acceleration peaks, and increased irregularities[2,9,11,12]. Recent evidence suggests that STN-DBS can improve several drawing parameters including the smoothness of movement, tremor, and a compound score termed degree of severity[13]. However, the relationship between drawing impairments, or potential DBS-related alleviations of such, with oscillatory neural activity within the STN remain to be examined. In the STN, increasing evidence shows pathologically exaggerated oscillatory synchronization in the beta frequency band (13–30 Hz) which correlates with motor impairment in PD[14–16]. Consistent findings demonstrate a movement-related desynchronization in beta power that is associated with multiple movement characteristics, highlighting its relevance for motor planning and execution[17–21]. It is well established that DBS reduces enhanced beta power, and that this effect is linked to the alleviation of motor impairments[22,23].

Beta oscillations come and go in transient bursts that can be determined on a patient-specific level, and increasing evidence highlights the relationship between bursts occurring prior to movement and subsequent movement characteristics[24–28]. Considering their high signal-to-noise ratio in combination with the simplicity and speed of analytical processing, beta bursts seem to be particularly practical to control the stimulation in the framework of closed-loop, adaptive DBS systems for patients with PD[29,30]. However, while most studies have so far investigated bursts that are present briefly before movement, the occurrence of bursts during movement and the potential implications for DBS have not been thoroughly investigated. In this regard, Lofredi et al.[31] showed that beta bursts are still present during prolonged, simple pronation-supination movements and that the time spent bursting correlates with velocity decrements. Similarly, Kehnemouyi et al.[32] showed that DBS-related reductions in average burst duration is associated with increases in average movement velocity during a flexion-extension task that captures bradykinesia. However, it remains unclear if bursts occur during prolonged and fine-controlled movements that require the continuous integration of feedback and dynamic adjustments of movements. Furthermore, while conventional DBS reduces the burst amplitude during rest[24], a similar relationship during task execution remains to be shown. The possibility to interfere with burst activities during the execution of a task would provide new opportunities to investigate motor execution and require new and hitherto neglected considerations for upcoming adaptive DBS-systems. An important question is whether a general reduction in burst amplitude affects fine motor control, or if targeting specific beta bursts (for example only prolonged bursts) is needed to improve fine-controlled movements in patients with PD.

Altogether, we aimed to investigate the functional relationship between the execution of a spiral drawing task and STN beta activities in patients with PD, and how this is affected by DBS. Because the presentation of visual or auditory stimuli modulates motor programs and these cues can alleviate parkinsonian symptoms[33,34], we designed a task where the participants were asked to draw spirals under two distinct conditions: drawing freely, which represents an internally guided movement, and drawing with a template that provides additional external visual cues. The two tasks require varying levels of accuracy and affect the contributions of sensory-visual feedback. The motivations for focusing on beta bursts was their relevance for movement and them being a major candidate for feedback signals for adaptive DBS. We asked three specific questions: (1) Are beta bursts present, albeit modulated, during drawing movements? (2) Is the drawing velocity affected by the occurrence of beta bursts? (3) Does continuous DBS interfere with bursts during drawing, and thus influence movement execution?

We hypothesized that the execution of drawing movements involves a modulation of beta burst activities. Moreover, we hypothesized that the occurrence of beta bursts directly affects movement execution. And finally, we examined the impact of STN-DBS on the duration, amplitude, and rate of beta bursts occurring during drawing to elucidate how continuous DBS interferes with STN beta burst dynamics and if this translates to alterations in fine motor control and movement execution. To this end, we recorded STN local field potentials (LFPs) via externalized electrode extension leads during a drawing task both with and without the application of high-frequency stimulation in patients briefly after DBS surgery.

## Results
Nineteen patients with PD who were externalized shortly after DBS electrode implantation performed the spiral-drawing task (Fig. 1). Participants drew spirals with their dominant hand both freely and guided by a template. We recorded LFPs directly from the STN during task execution via the externalized DBS electrodes. The task was performed with and without subthalamic electrical stimulation at 130 Hz. Two patients could not complete all recordings due to fatigue. The order was randomized and patients were blinded to the stimulation condition. Demographics and disease related information are summarized in Table 1 and individual clinical details are listed in Supplementary Table 1. Stimulation markedly improved the blinded UPDRS$_{III}$ scores (stim off = 25.16 (±8.74); stim on = 19.05 (±7.43); t = 6.374, $P < 0.001$, 95% CI = [4.09 8.12], Cohen's d = 0.75). Because the drawn lines crossed their own trace in some trials, we performed our analyses both with the complete data (free - stim_off, n = 167; free - stim_on, n = 161; template - stim_off, n = 179; template - stim_on, n = 180) and after excluding inaccurate trials (free - stim_off, n = 123; free - stim_on, n = 123; template - stim_off, n = 154; template - stim_on, n = 151). We also assessed the robustness of our analyses when excluding seven tremor-dominant patients.

### Deep brain stimulation increases drawing velocity
To evaluate graphomotor execution, we first calculated the average tangential velocity of each spiral drawing (Fig. 1A). Data were log-transformed due to positively skewed distributions. Before relating motor performance to STN activity, we compared the velocity between drawing and stimulation conditions (Fig. 2A). We found that patients drew the spirals significantly faster when they were not constrained by a template ($P < 0.001$). Similarly, delivering DBS via externalized leads significantly sped up drawing movements ($P = 0.003$). There was no evidence in favor of an interaction effect ($P = 0.135$). To assess the robustness of this result, we conducted several post hoc tests. First, we excluded trials where the pen crossed its own trace (defined as inaccurate trials). Both main effects remained significant ($P < 0.001$). Secondly, seven patients among our subjects had the tremor-dominant phenotype. To test if the inclusion of tremor-dominant patients affects our results, we repeated the analysis excluding them. Again, both main effects remained significant (effect of drawing condition: $P < 0.001$; effect of stimulation: $P = 0.004$).

Next, we tested if providing a template and applying stimulation affected spatial features of drawing. To this end, we calculated the radius-angle-transformation (Supplementary Fig. 1), fitted linear models for these transformations, and calculated their root-mean-square errors (RMSE) and slopes as parameters of deviation from an optimal spiral. The RMSE was increased when drawing freely ($P < 0.001$), but not affected by stimulation ($P = 0.051$). There was no evidence in favor of an interaction effect ($P = 0.059$). The main effect of

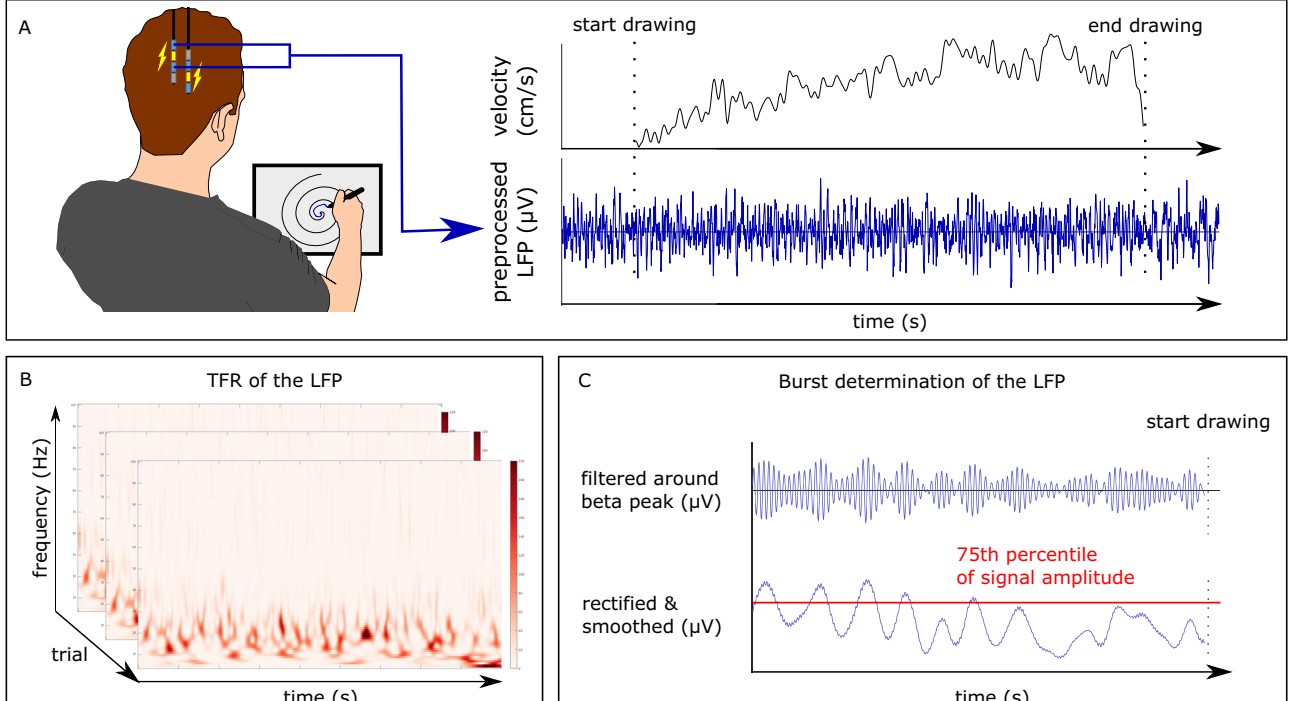

**Fig. 1 | Experimental setup and analytical steps. A** Patients drew spirals with their dominant hand on a digital tablet while we recorded local field potentials (LFP) from the bilateral subthalamic nuclei in a 'wide' bipolar montage. This allowed us to deliver deep brain stimulation simultaneously from the interleaved contact. The task was performed with and without stimulation, in randomized order. An example of the tangential velocity and the preprocessed LFP (high- and lowpass-filter at 4 Hz and 100 Hz; down-sampling; DFT-filter; demeaning and detrending) are presented as a function of time (right). After offline preprocessing the LFP signals were analyzed in two different steps. **B** LFP-signals were transformed to the time-frequency representation (TFR) from 4 Hz to 100 Hz with a frequency resolution of 1 Hz and 20 ms for the center of the moving window[21]. The Colorscale denotes the power within a time-frequency window. **C** The LFP signal was filtered around individually determined beta frequencies (Supplementary Table 1), rectified, and smoothed to obtain the envelope of the beta activity. For each condition, a threshold was then set at the 75th percentile of the beta amplitude of the corresponding rest interval. The onset of a burst was defined as when the rectified signal crossed the threshold amplitude while the end of the burst was defined as when the amplitude fell below the threshold. All bursts with a duration longer than 100 ms were considered.

drawing condition remained significant when excluding inaccurate trials and when excluding tremor patients (both $P < 0.001$).

The slope of the fitted model was reduced when drawing freely ($P < 0.001$), indicating that patients drew larger spirals when guided by a template. Furthermore, stimulation increased the slope ($P = 0.002$). Both effects remained significant when excluding inaccurate trials (both $P < 0.001$) as well as tremor-dominant patients (effect of drawing condition: $p < 0.001$; effect of stimulation: $P = 0.031$).

To test if spatial parameters influenced the velocity, we separately added the RMSE and the slope as fixed factors to our original model. While both RMSE and the slope were positively associated with the velocity (both $P < 0.001$), the main effects of drawing and stimulation condition remained significant (effect of drawing condition: both $P < 0.001$ when controlling for RMSE and slope, respectively; effect of stimulation: both $P = 0.04$ when controlling for RMSE and slope, respectively). There were no interactions between stimulation and drawing condition. These results were not affected by excluding inaccurate trials (effect of drawing condition: both $P < 0.001$ when controlling for RMSE and slope, respectively; effect of stimulation: $P = 0.003$ and $P = 0.039$ when controlling for RMSE and slope, respectively). However, the effect of stimulation did not remain significant when excluding tremor-dominant patients and controlling for RMSE ($P = 0.055$). The effects of drawing condition were not affected when excluding tremor-dominant patients (both $P < 0.001$ when controlling for RMSE and slope, respectively).

These results show that patients generally drew slower when a template was provided, which could be attributed to an increase of accuracy constraints. Applying DBS in the STN generally sped up drawing movements.

## Reduced beta power during drawing in the subthalamic nucleus
During the task, we recorded LFPs directly from the STN via temporarily externalized DBS electrodes. This allowed us to assess the electrophysiological response of STN activities within the beta

**Table 1 | Patient characteristics**

| | Patients with Parkinson's disease |
|---|---|
| Number of participants | 19 |
| Age | 67.68 (7.25; 49–80) |
| Sex (male/female)[a] | 15/4 |
| Disease duration (years) | 10.42 (4.82; 3–22) |
| Hoehn & Yahr | 3.24 (0.77; 2–5) |
| UPDRS$_{III}$ pre surgery (off levodopa) | 34.16 (14.45; 8–62) |
| UPDRS$_{III}$ pre surgery (on levodopa) | 23.47 (13.18; 5–54) |
| LEDD | 1212.42 (521.52; 440–2128) |
| DBS intensity contralateral (mA)[b] | 2.242 (0.995; 1–5) |
| DBS intensity ipsilateral (mA)[b] | 2.211 (1.080; 1–5) |
| Order of stimulation (off-on/on-off) | 10/9 |
| UPDRS$_{III}$ (DBS off/DBS on)[c] | 25.16 (8.74; 10–39)/19.05 (7.43; 4–29)[d] |

Clinical scores are given as total score of the Movement Disorder Society Unified Parkinson's disease rating scale (UPDRS) part III for levodopa off/on (pre-surgery) and as items 3–8 & 14–18 (limb scores) for DBS off/on. Medication is given in levodopa-equivalent daily dose (LEDD). Standard deviation and range are presented within the brackets.
[a]Sex was assigned.
[b]Stimulation intensity for electrodes of the contralateral and ipsilateral STN (in relation to the drawing hand) in mA.
[c]Items 3.3, 3.4, 3.5, 3.6, 3.7, 3.8, 3.14, 3.15, 3.16, 3.17, and 3.18.
[d]$P < 0.001$ (two-sided paired samples t-test).

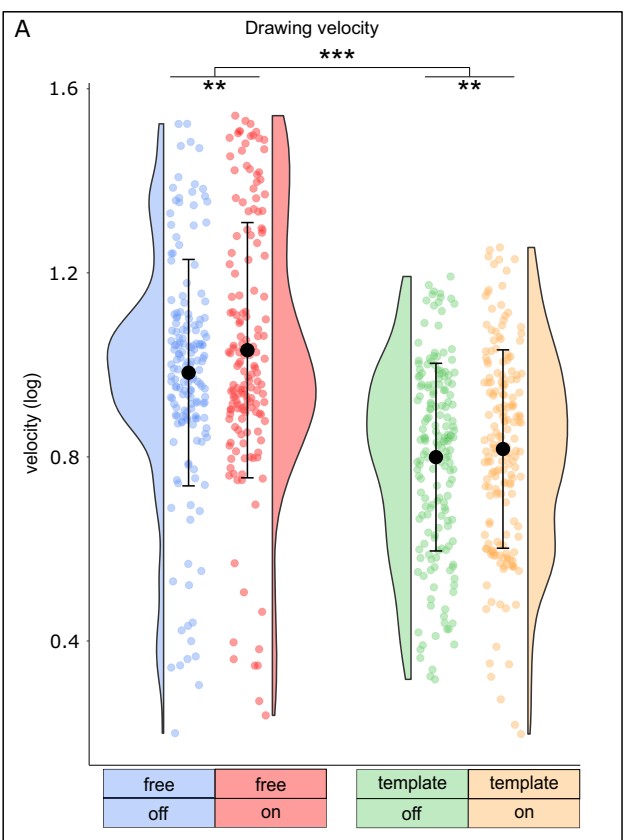

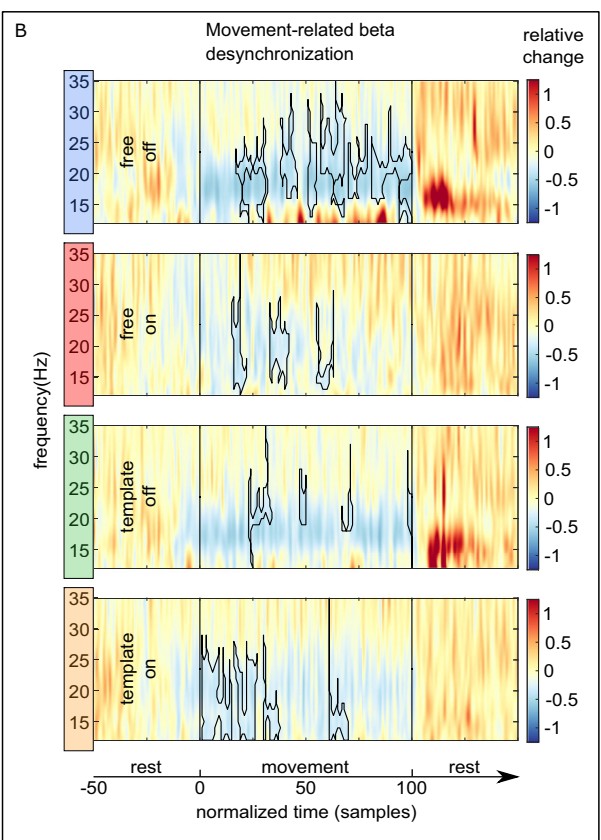

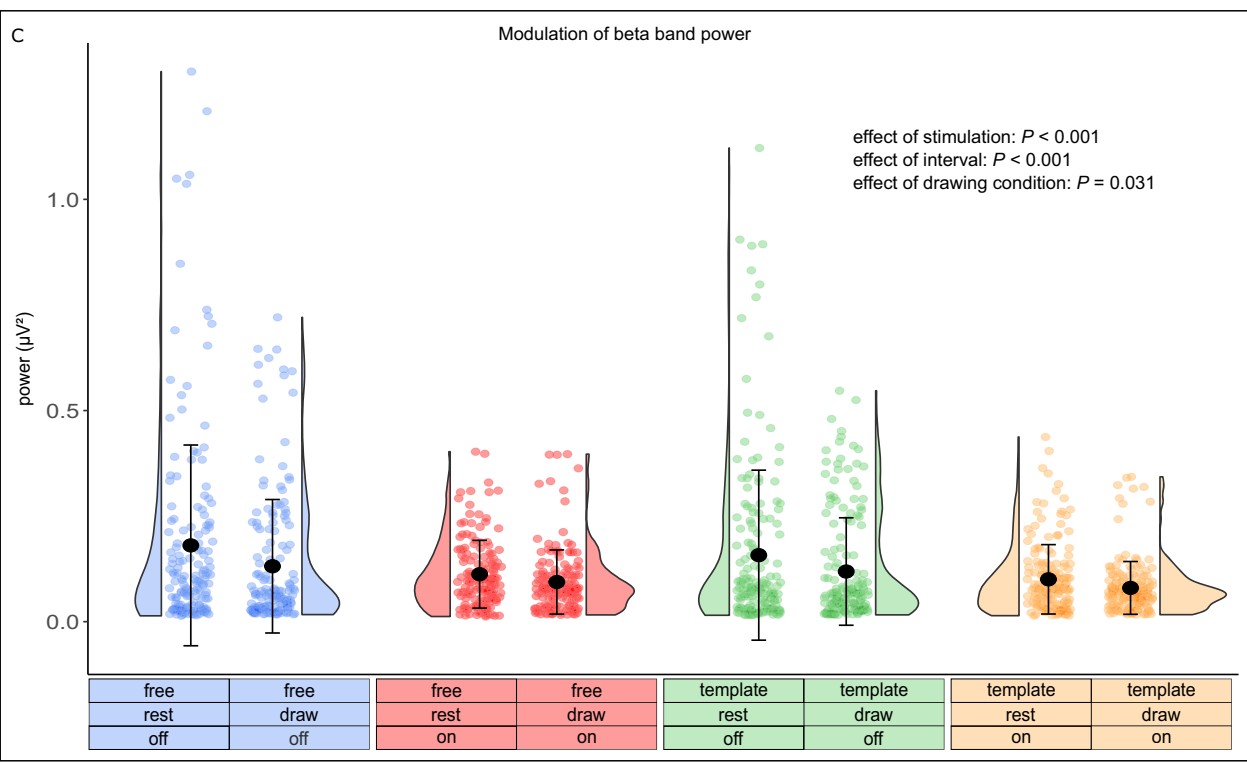

frequency band associated with complex and continuous movements. Due to the variable duration of the trials, we temporarily normalized the LFP-data before performing cluster based permutation tests. Figure 2B displays the desynchronization in broadband beta frequencies that lasted from the start of the movement until its termination. Cluster based permutation tests revealed significant clusters ($P < 0.05$,

one-sided) in both the early and late drawing intervals. The exact cluster statistics are shown in Supplementary Table 2.

To assess if movement and stimulation also affects broadband beta and gamma power we performed a frequency analysis for the rest and drawing intervals and averaged the power of frequencies between 13 Hz and 30 Hz (beta) and 30 Hz and 45 Hz (gamma) within each trial.

**Fig. 2 | Graphomotor performance and drawing related beta band desynchronization. A** Individual trials, distribution, mean, and standard deviation of the average drawing velocity (log transformed) for all four conditions (blue = free drawing without stimulation: 0.984 arbitrary units (arb. units) ±0.247 (mean ± standard deviation), $n = 167$; red = free drawing with stimulation: 1.033 arb. units ± 0.279, $n = 161$; green = template-guided drawing without stimulation: 0.800 arb. units ± 0.205, $n = 179$; orange = template-guided drawing with stimulation: 0.818 arb. units ± 0.216, $n = 180$). There was a significant effect of drawing ($P < 0.001$, two-sided linear mixed-effects model) and stimulation condition ($P = 0.003$, two-sided linear mixed-effects model), indicating that patients drew faster when no template was presented as well as when stimulation was delivered. **B** Grand average time-frequency plots of the LFPs from the contralateral STN show a reduction of beta activities in the range of ~15–25 Hz during the drawing period. Significant clusters ($P < 0.05$, one-sided cluster based permutation test) are encircled by the black borders. Vertical black bars indicate the beginning (at sample 0) and end (at sample 100) of drawing. Color-bars represent the relative increase/decrease in spectral power normalized by the baseline interval (fieldtrip function ft_singleplotTFR with cfg.baselinetype = 'relchange'). **C** Individual values, distribution, mean, and standard deviation of the broad-band beta power for all four conditions (blue = free drawing without stimulation, draw: 0.131 microvolt ($\mu V^2$) ± 0.158 (mean ± standard deviation), $n = 165$, rest: 0.180 $\mu V^2$ ± 0.238, $n = 165$; red = free drawing with stimulation, draw: 0.094 $\mu V^2$ ± 0.076, $n = 156$, rest: 0.112 $\mu V^2$ ± 0.080, $n = 156$; green = template-guided drawing without stimulation, draw: 0.119 $\mu V^2$ ± 0.127, $n = 173$, rest: 0.157 $\mu V^2$ ± 0.201, $n = 173$; orange = template-guided drawing with stimulation, draw: 0.080 $\mu V^2$ ± 0.063, $n = 175$, rest: 0.100 $\mu V^2$ ± 0.082, $n = 175$). Two-sided linear mixed-effects models showed that there were significant main effects of stimulation condition ($P < 0.001$), drawing condition ($P = 0.031$), and movement interval ($P < 0.001$), indicating that broadband beta power was reduced by stimulation, movement, and during template-guided drawing in comparison to free drawing. **$P < 0.01$; ***$P < 0.001$.

We tested the effects of stimulation (off vs on), movement interval (rest vs draw), drawing condition (free vs template), as well as all possible interactions with linear mixed-effects models. With respect to beta power (Fig. 2C), this showed significant main effects for stimulation condition ($P < 0.001$), movement interval ($P < 0.001$), drawing condition ($P = 0.031$), and an interaction between stimulation and movement interval ($P = 0.030$). Post hoc tests revealed that the interaction effect was based on a relative reduction of beta power in the rest interval during stimulation in comparison to drawing without stimulation ($P = 0.021$). Otherwise, post hoc tests confirmed that stimulation generally reduced beta power both in the rest as well the drawing intervals (both $P < 0.001$), and that drawing reduced broadband beta power, independent of the stimulation condition (off: $P < 0.001$, on: $P = 0.035$). With respect to gamma power, we did not identify any significant effects (all $P > 0.05$).

To assess a potential association between the drawing velocity and the extent of movement-related desynchronization, we performed an additional linear model with velocity as the dependent variable, stimulation condition, drawing condition, the difference between beta power during drawing and rest, as well as the interactions between these three parameters as fixed effects. This showed an interaction between the drawing condition and the movement related desynchronization ($P = 0.010$). However, post hoc tests did not show any association in either drawing condition (both $P > 0.05$).

## Beta bursts are modulated during drawing

Our first question was whether beta bursts are modulated during the execution of a drawing task. Therefore, we calculated the number of bursts, burst rate, mean burst duration, and mean burst amplitude for the rest and drawing intervals for all conditions. We tested the effects of stimulation (off vs on), movement interval (rest vs draw), drawing condition (free vs template), as well as all possible interactions on the beta burst characteristics. We applied linear mixed-effects models with burst duration, burst amplitude, number of bursts, and burst rate as the dependent variables. Detailed results of the applied post hoc tests can be seen in Supplementary Fig. 2.

Regarding the mean burst duration (Fig. 3), we found a significant effect of movement interval ($P < 0.001$), showing that drawing reduced the duration of beta bursts in comparison to the rest interval. Additionally, we found a significant interaction between stimulation and movement interval ($P = 0.039$). However, the post hoc tests revealed that the drawing intervals were consistently characterized by shorter burst durations in comparison to the rest intervals, regardless of the stimulation condition ($P < 0.001$). Furthermore, applying DBS did not significantly affect burst duration in either the rest ($P = 0.638$) or the drawing intervals ($P = 1$). There were no main effects of stimulation ($P = 0.355$) or drawing condition ($P = 0.646$). For better comparison with similar studies that quantified beta burst characteristics, Supplementary Fig. 3 presents the data without log-transformation.

We additionally conducted post hoc tests analogously to the analyses of the drawing parameters. Excluding inaccurate trials did not alter the main effect of drawing interval ($P < 0.001$) but the interaction effect ($P = 0.130$). Excluding seven tremor-dominant patients did not alter the main effect of drawing interval ($P < 0.001$), but the interaction effect ($P = 0.098$).

Regarding the mean burst amplitude (Fig. 4), we found significant effects of stimulation ($P < 0.001$) and movement interval ($P < 0.001$), showing that stimulation and drawing reduced the beta burst amplitude. There was no difference in respect to the drawing condition ($P = 0.076$). Additionally, we found a significant interaction between stimulation and movement interval ($P = 0.018$). The post hoc tests revealed that the mean burst amplitude was reduced during stimulation both during the rest ($P < 0.001$) and drawing intervals ($P = 0.012$). Furthermore, the mean burst amplitude was reduced in the drawing interval in comparison to the rest interval without stimulation ($P < 0.001$), but not to the rest interval with stimulation ($P > 0.05$). For better comparison with similar studies that quantified beta burst characteristics, Supplementary Fig. 4 presents the data without log-transformation.

We additionally conducted post hoc tests analogously to the analyses of the drawing parameters. When excluding inaccurate trials, the main effects of movement interval ($P < 0.001$) and stimulation ($P < 0.001$) remained significant. However, there was no interaction between stimulation and the drawing interval ($P = 0.052$). Excluding seven tremor-dominant patients did not alter the main effects of movement interval ($P < 0.001$), stimulation condition ($P < 0.001$), nor the interaction between stimulation and movement interval ($P = 0.034$).

Regarding the number of bursts (Supplementary Fig. 5), we found a significant effect of movement interval ($P = 0.002$) and a significant interaction between movement interval and drawing condition ($P < 0.001$). The post hoc tests revealed that the drawing interval was characterized by an increased number of bursts ($P < 0.001$), and the number of bursts was even higher during drawing on a template in comparison to free drawing ($P < 0.001$). Therefore, because these drawing intervals were longer in comparison to the resting intervals that were fixed at 5 s, the absolute number of bursts increased during movement execution. As expected, the number of bursts were not significantly different between the rest intervals ($P = 1$). There were no main effects of stimulation ($P = 0.541$) or drawing condition ($P = 0.775$).

We additionally conducted post hoc tests analogously to the analyses of the drawing parameters. When excluding inaccurate trials or tremor-dominant patients, the main effects of movement interval (both $P < 0.001$) and the interaction between movement interval and drawing condition (both $P < 0.001$) remained significant.

Regarding the burst rate we found a significant effect of interval ($P < 0.001$) showing that drawing reduced the beta burst rate in comparison to rest. No further main effects or interactions were found. We

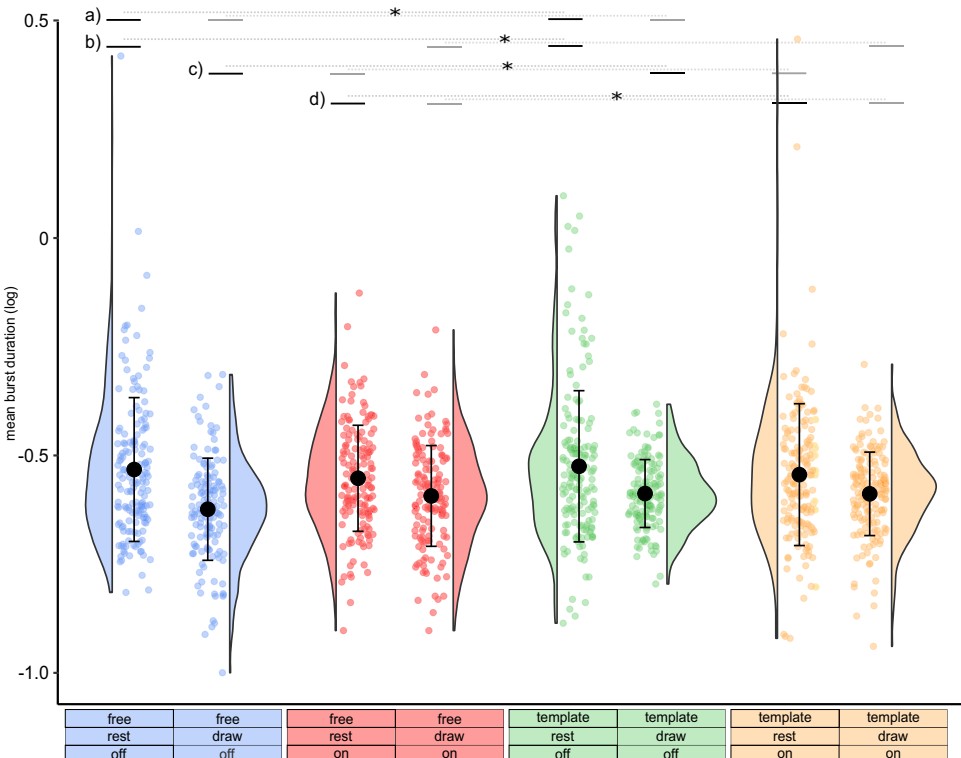

**Fig. 3 | Burst duration is reduced during drawing in comparison to the rest interval.** Single trials, distributions, mean and standard deviation of the mean burst duration are plotted for the different conditions and intervals (blue = free drawing without stimulation, draw: −0.624 arbitrary units (arb. units) ±0.117 (mean ± standard deviation), $n = 149$, rest: −0.533 arb. units ±0.165, $n = 164$; red = free drawing with stimulation, draw: −0.593 arb. units ± 0.116, $n = 153$, rest: −0.553 arb. units ± 0.122, $n = 156$; green = template-guided drawing without stimulation, draw: −0.588 arb. units ±0.078, $n = 157$, rest: −0.525 arb. units ±0.174, $n = 172$; orange =

template-guided drawing with stimulation, draw: −0.589 arb. units ±0.096, $n = 167$, rest: −0.544 arb. units ±0.163, $n = 173$). Two-sided linear mixed-effects model revealed a significant effect of movement interval ($P < 0.001$) as well as an inter-action between stimulation and movement interval ($P = 0.039$). The top lines indicate the results from the significant post hoc tests (two-sided, bonferroni corrected), combined across the drawing conditions (free and template): **a** rest_off > draw_off, $P < 0.001$; **b** rest_off > draw_on, $P < 0.001$; **c** rest_on > draw_off, $P < 0.001$; **d** rest_on > draw_on, $P = 0.001$. *$P < 0.05$ (Bonferroni corrected).

additionally conducted post hoc tests analogously to the analyses of the drawing parameters. When excluding inaccurate trials or tremor-dominant patients, the main effect of movement interval remained significant ($P < 0.001$ and $P = 0.016$, respectively).

These results show that bursts are still present, yet modulated during drawing. When a template was provided, patients generally drew slower. Applying DBS in the STN sped up drawing movements and generally reduced burst amplitude, but not burst duration.

## Beta bursts affect velocity dynamics during drawing

Our second question was if the velocity of drawing movements is affected by the occurrence of beta bursts. To address this, we evaluated the dynamics of the velocity signals in the time-periods shortly before and after the beginning of a burst (Fig. 5A). Because the velocity generally increased during task execution due to the increase of the spiral's diameter, we calculated the average acceleration within 250 ms intervals before and after each burst. Then we tested if the acceleration differed between the two intervals, and if it was affected by the stimulation or drawing conditions. We found significant main effects of the time interval ($P < 0.001$) and the stimulation condition ($P = 0.001$), but not the drawing condition ($P = 0.415$). We further found interaction effects for stimulation condition * drawing condition ($P = 0.007$) and time interval * drawing condition ($P = 0.019$). Following the two two-way interactions, the post hoc tests showed that the effects of time interval and stimulation condition were only present during free drawing ($P = 0.002$ and $P = 0.008$, respectively).

The results were not different when excluding tremor-dominant patients (both main effects and both interaction effects remained at

$P < 0.05$). Apart from the interaction between time interval * drawing condition ($P = 0.057$), the results were similar when excluding inaccurate trials (main effects of stimulation and drawing condition and interaction between stimulation condition * drawing condition: all $P < 0.05$). To account for the potential influence of preceding bursting activities on the velocity of movement[26], we additionally tested if including the burst duration, burst amplitude, and burst rate of the baseline interval as fixed factors changes the results, which was not the case (both main effects and both interaction effects remained at $P < 0.05$). Furthermore, neither of these parameters had an effect on the acceleration (all $P > 0.05$).

To investigate if the occurrence of a burst not only accompanies the velocity dynamics but also deviations from the optimal spiral trace, we extracted the residuals of the fitted models of each spirals' radius-angle transformation. Similarly to the evaluation of the velocity dynamics, we tested if the instantaneous deviation from the optimal trace differed between the two intervals, and if it was affected by the stimulation or drawing conditions. We did not find any main effects or any interaction effects (all $P > 0.05$), showing that the occurrence of a burst does not directly affect deviations from an optimal drawing trace. The results were not different when controlling for bursting activities in the baseline interval (all $P > 0.05$).

Because participants executed the task with and without the application of clinically effective high-frequency stimulation, we were able to test if DBS-related modulations of beta bursts during drawing influenced movement execution (question 3). Specifically, we applied linear mixed-effects models to test if amplitude or duration of beta bursts are associated with the extent of the acceleration alterations.

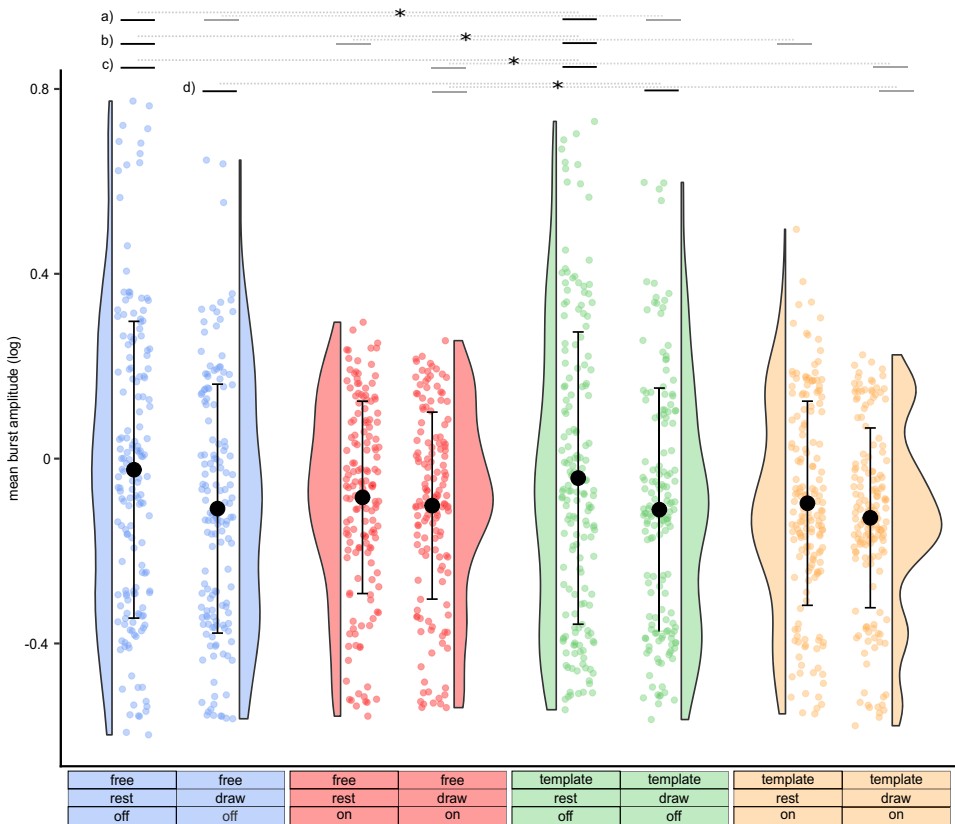

**Fig. 4 | Burst amplitude is reduced during drawing in comparison to the rest interval and reduced by deep brain stimulation.** Single trials, distributions, mean and standard deviation of the mean burst amplitude are plotted for the different conditions and intervals (blue = free drawing without stimulation, draw: −0.108 arbitrary units (arb. units) ±0.269 (mean ± standard deviation), n = 149, rest: −0.024 arb. units ±0.321, n = 164; red = free drawing with stimulation, draw: −0.102 arb. units ±0.202, n = 153, rest: −0.084 arb. units ±0.208, n = 156; green = template-guided drawing without stimulation, draw: −0.110 arb. units ±0.263, n = 157, rest: −0.042 arb. units ±0.316, n = 172; orange = template-guided drawing with stimulation, draw: −0.128 arb. units ±0.195, n = 167, rest: −0.097 arb. units ±0.221, n = 173). Two-sided linear mixed-effects model revealed significant effects of movement interval (P < 0.001), stimulation (P < 0.001), as well as a significant interaction between stimulation and movement interval (P = 0.018). The top lines indicate the results from the significant post hoc tests (two-sided, bonferroni corrected), combined across the drawing conditions (free and template): **a** rest_off > draw_off, P < 0.001; **b** rest_off > rest_on, P < 0.001; **c** rest_off > draw_on, P < 0.001; **d** draw_off > draw_on, P < 0.0012. *P < 0.05 (Bonferroni corrected). Source data are provided as a Source Data file.

We found significant interactions between stimulation and burst amplitude (P = 0.009) and between stimulation, drawing condition and burst amplitude (P = 0.036), suggesting that the association between burst amplitude and acceleration alterations is more negative when patients receive stimulation, and also affected by the drawing condition. However, separate linear mixed-effects models for the drawing and stimulation conditions did not show any significant association between the alteration in acceleration and the beta burst amplitude (all P > 0.05). When excluding inaccurate trials, both inter-actions were not significant (P > 0.05). In contrast, both interactions remained significant when excluding tremor-dominant patients (P = 0.001 and P = 0.02, respectively). Lastly, the results remained similar when controlling for bursting activities in the baseline interval. Interestingly, the average burst duration before drawing was generally associated with the extent of acceleration reductions following a single burst (P = 0.045).

In the second linear mixed-effects model testing if beta burst duration of bursts occurring during drawing is associated with the extent of acceleration alterations, we did not find any significant main effects of burst duration or any interactions between stimulation, drawing condition, and burst duration (all P > 0.05). This was not affected by adding preceding burst activities as fixed effects in the model.

Together, these results demonstrate that beta bursts parallel reductions in acceleration when drawing freely. Additionally, the implementation of DBS specifically modifies movement acceleration around beta bursts in this drawing context.

## Reduced beta burst amplitude correlates with clinical improvement

We then investigated a potential relationship between the DBS-related beta burst amplitude reduction and clinical improvement. Patients with stronger DBS-related reductions in burst amplitude presented increased clinical improvement: as expected, reductions of the amplitude of bursts occurring in the rest intervals of both drawing conditions were associated with improved contralateral arm-scores (rest free drawing: Spearman rho = 0.632, P = 0.026; rest template-guided drawing: Spearman rho = 0.625; P = 0.017). Furthermore, reductions in amplitude during drawing were also correlated with clinical improvement (free drawing: Spearman rho = 0.634, P = 0.033; template-guided drawing: Spearman rho = 0.614; P = 0.027). Exemplary scatter plots demonstrating the correlations are shown in Fig. 6 and the detailed results are presented in Supplementary Table 3.

Five participants showed only minor or no clinical improvement of total UPDRS$_{III}$ when receiving DBS during the study. This might be related to the brief time interval between study participation and the DBS surgery, where micro-lesions might improve motor function even in the absence of stimulation[35]. Such lesions might have induced a temporary ceiling effect, where stimulation does not further improve motor function. Another explanation could be the relatively short

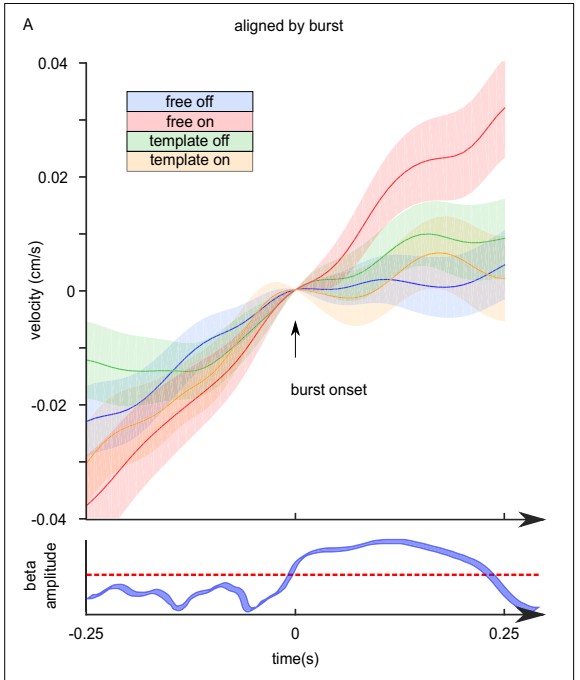

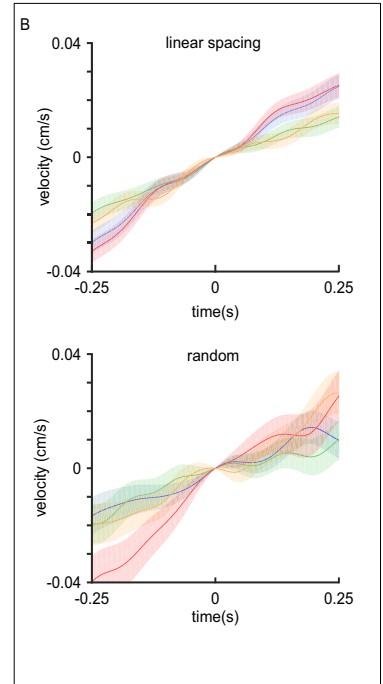

**Fig. 5 | Deep brain stimulation alleviates burst related reductions of acceleration during free drawing. A** Top: For visualization, velocity signals within windows of 250 ms before and after the occurrence of a burst were normalized by the instantaneous velocity during burst onset and averaged within each subject. Average velocity signals and the standard error of the mean (shaded area) for all drawing and stimulation conditions (blue = free drawing without stimulation; red = free drawing with stimulation; green = template-guided drawing without stimulation; orange = template-guided drawing with stimulation) are shown. We calculated the average acceleration within the pre and post intervals for each burst. Two-sided linear mixed-effects models showed main effects of time interval ($P < 0.001$), and stimulation condition ($P = 0.001$) and interaction effects for stimulation condition * drawing condition ($P = 0.007$), and drawing condition *time interval ($P = 0.019$). Two-sided post hoc tests demonstrated that only bursts occurring during free drawing are accompanied by a reduction in acceleration (as can be seen as a flattening of the blue velocity signal after the occurrence of a burst, $P = 0.002$). When applying DBS, the acceleration around beta bursts was generally increased during free drawing, suggesting that stimulation modifies their immediate impact on velocity dynamics ($P = 0.008$, two-sided post hoc). When drawing with a template, acceleration was not affected by the occurrence of a burst ($P = 1$, two-sided post hoc). Bottom: modeled beta amplitude dynamics demonstrating the definition of the burst onset as the time point when the amplitude crosses the 75th percentile threshold. **B** Average velocity dynamics and the standard error of the mean (shaded area) for the two control conditions showing that the velocity signal does not change its slope after the occurrence of linearly or randomly defined onsets (top: burst onsets defined in 50 ms steps across every trial; bottom: randomly assigned timestamps). Two-sided linear mixed-effects models did not show any significant effects of time interval ($P = 0.143$ and $P = 0.256$, linear and random assignment, respectively), stimulation condition ($P = 0.456$ and $P = 0.193$) or interaction between stimulation condition * time interval ($P = 0.913$ and $P = 0.867$), highlighting that the average acceleration after the defined onsets is not significantly different from before. Source data are provided as a Source Data file.

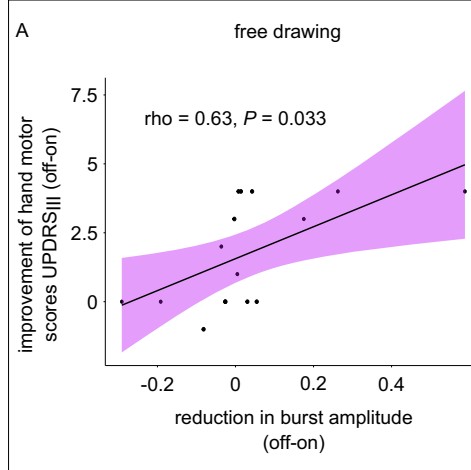

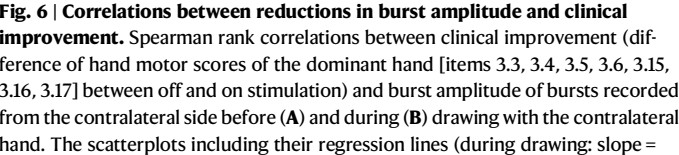

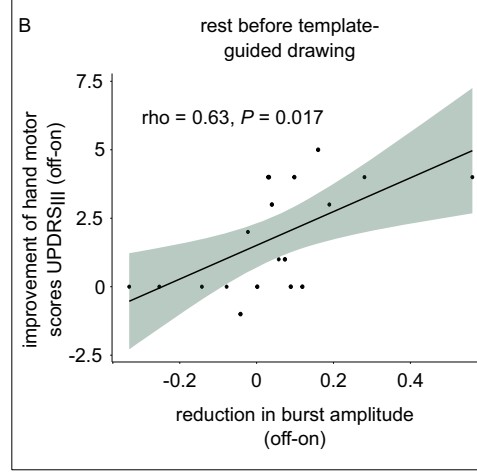

**Fig. 6 | Correlations between reductions in burst amplitude and clinical improvement.** Spearman rank correlations between clinical improvement (difference of hand motor scores of the dominant hand [items 3.3, 3.4, 3.5, 3.6, 3.15, 3.16, 3.17] between off and on stimulation) and burst amplitude of bursts recorded from the contralateral side before (**A**) and during (**B**) drawing with the contralateral hand. The scatterplots including their regression lines (during drawing: slope = 5.792, intercept = 1.560, $n = 16$ before drawing: slope = 6.160, intercept = 1.510, $n = 19$) and their 95% confidence intervals (shaded areas) show that patients that display increased reductions of burst amplitude, both in the rest as well as in the drawing intervals, exhibit increased motor improvement. *P*-values were Bonferroni-corrected. Source data are provided as a Source Data file.

wash-in and wash-out periods in between conditions. Finally, the electrode position might not have been optimal in these patients[36]. In two of these five patients, subclinical effects congruent with our behavioral findings could still be detected when being stimulated. In two patients, drawing performance was not affected and in one patient drawing velocity was reduced. The detailed results are presented in Supplementary Table 4 and the electrode locations modeled by Lead-DBS (Version 2.5.2, https://www.lead-dbs.org/)[37] are presented in Supplementary Fig. 6.

## Discussion

How do we dynamically control and adjust precise and prolonged movements? And why is the execution of such movements impaired in PD? Beta activity in the STN has been suggested to regulate motor planning and movement execution[17,19–21], and specifically the timing of movement in relation to the preceding beta bursts relates to the movement velocity[26,38]. In PD, beta bursts are prolonged, and the burst amplitude is increased, which is associated with motor impairment[16,24,28,31]. In this study we reveal a more diverse and complex picture of the relationship between subthalamic beta bursts and movement velocity: We show that the occurrence of beta bursts during the execution of free drawing movements is accompanied by an immediate reduction in velocity, which is not the case when movement is restricted by the presence of a guiding template. While both free and restricted drawing require fine and skilled hand- and finger-coordination, this suggests that increased accuracy constraints posed by the template diminish or mask the impact of pathologically exaggerated beta bursts. Delivering high-frequency stimulation to the STN increased the acceleration around beta bursts, also exclusively during free drawing. Considering such context-dependent modulations of beta bursts should help to optimize and enhance future DBS systems.

To investigate the functional relationship between the execution of a drawing task and STN beta burst dynamics, we first addressed the question whether the execution of drawing movements affects the characteristics of beta bursts that were defined during rest. Despite the overall suppression of beta oscillatory activities, bursts that crossed resting beta thresholds could still be detected during drawing. However, duration, amplitude, and burst rate were reduced in comparison to the resting interval, demonstrating that task-related processing is characterized by modulations of bursting activities. To investigate the functional relevance of these findings, we then asked if the velocity of drawing movements is directly affected by the occurrence of beta bursts. Our results demonstrate that bursts occurring during free drawing were accompanied by reductions in acceleration such that the velocity was slowed down afterwards. Notably, due to the increase of the spiral diameter, the drawing velocity generally increases during task execution (see for example, Fig. 1A). Thus, the reduced acceleration represents a reduction in the rate with which the velocity increases, which seems to be aligned with the occurrence of STN beta bursts. Although these results suggest that the occurrence of beta bursts immediately relates to the velocity dynamics of free drawing movements, we did not find a link between the extent of decreased acceleration and neither the burst amplitude nor burst duration. In contrast, we found that the average beta burst duration in the rest interval before drawing generally affected the extent of acceleration changes. As such, the recent cumulative history of beta bursts before motor execution could influence immediate alterations induced by an individual burst during fast movements. It remains open if preceding and recent activities within other frequencies might also impact an individual burst's ramifications.

Intriguingly, bursts occurring during template-guided drawing did not affect the velocity dynamics in a similar way. While both free and template-guided drawing require fine and skilled hand- and finger-coordination, one difference is that the presence of guiding lines poses an additional accuracy constraint for a patient. Accordingly, patients reduced the drawing velocity in comparison to free drawing, potentially to integrate visual feedback and exert active control to flexibly adjust the pen-movements. Supporting the presence of such a speed-accuracy trade-off, our results demonstrate that the presentation of the template was accompanied by reductions in deviations from an optimal trace and increases in the spiral's diameter. Importantly, RMSE and slope were directly associated with the drawing velocity. Consequently, the general reduction in movement velocity might mask the effects of beta bursts because pathological decrements are relatively small in relation to the average velocity and thus lose their relevance in highly controlled and slow movements. Furthermore, beta activity might play a different or additional role during feedback-guided in comparison to free movements. For example, it was shown that the integration of visual feedback modulates both cortical and subcortical beta oscillations, indicating that the regulation of beta activity also reflects neural processes related to gating feedback and evaluating movement deviations[39–41]. Consequently, an increased reliance on visual feedback posed by the template might modulate the role of beta bursts and thus influence the impact of single bursts on the velocity dynamics. Additionally, free drawing represents an internally guided movement, while presenting a template provides an external visual cue. Importantly, people with PD have particular difficulties with internally guided movements, and the presentation of visual or auditory cues can help patients to overcome movement impairments[33,34]. Externally guided movements are thought to preferentially involve cerebellar, premotor, and sensory circuits, while the basal ganglia and supplementary motor area (SMA), which are heavily affected by dopamine deficiency, play a more important role for internally guided movements[42–45]. The shift towards potentially less impaired motor circuits has been hypothesized to underlie the positive effects of external cueing strategies for addressing gait impairments in PD[34]. Likewise, presenting a template might engage supplementary pathways such as the cerebello-thalamo-cortical circuit in parallel, thereby influencing or surpassing the role of basal-ganglia beta oscillations. However, when comparing neural activation during tracing and free drawing movements with functional magnetic resonance imaging, the predicted shift in activation patterns could be shown in the pre-SMA but not in the cerebellum[44]. Thus, more research is needed to investigate temporo-spatial differences in electrophysiological activities between free and guided drawing movements. Taken together, these results highlight the relationship between the occurrence of bursts with the slowing of movement execution during free drawing, which fits the overall assumption that pathologically exaggerated bursts play a significant role for bradykinesia in PD[24,38]. The velocity of prolonged and slow, yet externally cued movements, on the other hand, might be relatively unaffected by beta bursts.

It was previously shown that both DBS and levodopa-treatment modulate beta bursting, suggesting that beta bursts are pathologically exaggerated and related to motor impairment in PD[24,31,46]. Thus, our third question was whether DBS interferes with beta bursts during drawing and may thereby mediate alterations in movement execution. As expected, patients drew faster when receiving DBS. While stimulation concurrently increased the spiral's radii, deviations from an optimal trace were not affected. Furthermore, stimulation reduced the burst amplitude both during task execution and in the rest interval, but did not affect burst duration. These results extend previous findings that suggest that burst amplitude is reduced by conventional DBS during rest[24], and suggest that bursts occurring during task execution are targeted in a similar manner. Despite the general reduction in beta burst amplitude, movement was still accompanied by visible modulations of burst duration and burst amplitude when receiving stimulation, highlighting that continuous DBS does not suppress beta activities entirely. This is important because a complete suppression might be just as detrimental for physiological functioning as an over-expression. When looking at the mechanisms of beta-burst triggered

adaptive DBS, for example, it was hypothesized that short bursts could be physiological and functionally relevant, while specifically prolonged bursts would impair motor function and increase over-all broad-band beta power[24]. The modulatory effect of continuous subthalamic stimulation on burst duration is less clear. In line with our results, Tinkhauser et al.[24] showed that conventional DBS does not alter burst duration during rest. Using a different approach for burst detection and a rigorous withdrawal of dopaminergic medication, Kehnemouyi et al.[32] showed that DBS reduces burst duration during repetitive flexion-extension movements of the wrist in chronically stimulated PD patients. Thus, it is possible that residual effects of medication or a surgery-related temporal lesioning effect[35] masks the modulation of beta burst duration. Kehnemouyi et al.[32] also demonstrated that the DBS-related reduction in average burst duration is associated with increases in average movement velocity during repetitive flexion and extension of the wrist when modulating the stimulation intensity. Expanding on their approach, we here investigated how subthalamic stimulation with an optimal intensity affects the specific relationship between single beta bursts and their immediate effects on the velocity dynamics, as presented during free drawing. In this context, DBS increased the acceleration around beta bursts, modifying their immediate ramifications on velocity dynamics during drawing in PD. The modulation of such relationship under stimulation corroborates the notion that DBS-related modulations of beta bursts might underlie the improvement of the progressive deterioration of velocity during ongoing movement in chronically implanted patients[47]. However, the acute intake of levodopa also reduces duration, amplitude, and rate of subthalamic beta bursts[46], and it remains to be elucidated how the effect of DBS might be modulated by levodopa.

Patients also increased the average drawing velocity during template-guided drawing when receiving DBS, although bursts did not present the same immediate effects on the velocity dynamics as during free drawing. Considering that template-guided drawing is an externally guided movement that likely engages cerebello-thalamo-cortical networks[42–44], DBS-related modulations of communication between STN and (pre-)SMA[48,49] can be expected to have a more pronounced impact on free, internally guided drawing movements that predominantly rely on basal ganglia-cortical networks. While our results demonstrate that DBS increased drawing velocity and decreased burst amplitude in both conditions, at least one additional mechanisms must play a role. Recent evidence suggests that average power and burst-characteristics can provide complementary information[50]. Nevertheless, we did not find any associations between the extent of movement-related desynchronization in the broadband beta range and the drawing velocity. Further investigation is necessary to determine whether the modulation of separate yet inter-connected pathways engaged in template-guided drawing contribute to a stimulation-related increase in drawing speed. Furthermore, the cumulative history of beta bursting briefly before motor execution was recently linked to slowing in patients with PD[26]. Consequently, additional increases in drawing velocity may in turn relate to a cumulative effect of continuous DBS on burst characteristics beyond the modulation of the relationship between specific bursts and the subsequent velocity dynamics. This could be an additional explanation for why conventional DBS can require several minutes to be fully effective after being switched on[23,51]. However, in the current study we did not find that bursts occurring before the drawing task influenced the general acceleration during drawing.

Together, the differential effects of bursts in relationship to our two tasks highlight the necessity to thoroughly investigate how different types of movement can be impacted by DBS. Our results suggest that bursts occurring during free drawing are paralleled by reductions in acceleration, a relationship that does not seem to be present during template-guided drawing. Furthermore, continuous DBS generally increases the acceleration around beta bursts during free drawing, thus ameliorating the impact of subthalamic beta bursts on the velocity dynamics. Such context-dependent modulation of burst activities needs to be considered in the development of future stimulation systems. For example, our results also highlight that stimulation generally increased the drawing velocity in both conditions, which could not be explained by the direct associations between beta bursts and velocity dynamics. Thus, additional factors are involved in determining movement speed[52], and it is conceivable that the context in which movement occurs determines the relevant mechanisms that can be targeted to optimally improve motor function.

Finally, we investigated if DBS-related reductions of beta burst amplitude related to clinical improvement. Expanding on previous findings that demonstrate a relationship between burst duration and clinical impairment[24], we found that those patients with higher DBS-related reductions in burst amplitude during rest presented increased improvements of their contralateral arm-scores. This highlights that burst properties not only relate to the current clinical impairment of patients, but that alterations in those characteristics within a patient are directly related to improvements of their clinical state. Similarly, we demonstrated that DBS-related reductions in amplitude during task execution correlated with clinical improvement. This suggests that not only the modulation of bursts that occur during rest, but also during the execution of motor tasks is relevant for the clinical improvement of PD patients treated with DBS. Especially in conjunction with the findings that bursts during task execution are similarly affected by stimulation as those that occur during rest, this, at least in conventional DBS, further corroborates a cumulative effect of stimulation-related reductions in burst amplitude.

In summation, our results suggest a more diverse and complex role of STN beta bursts for drawing movements performed by patients with PD, depending on specific task constraints. We demonstrate that bursts can be detected during drawing, where their occurrence is accompanied by velocity reductions when accuracy constraints are kept relatively low. This highlights the necessity to thoroughly study the role of beta bursts for movement execution and impairment in a context-dependent framework to optimize and enhance future DBS systems. Conventional DBS reduced the amplitude of bursts both in the rest interval and during task execution, and both immediate and cumulative effects might be relevant components mediating the stimulation-related increase in movement velocity in PD.

## Methods
### Participants and study protocol
We investigated the dynamics of beta bursts during a spiral-drawing task in 19 participants with PD who had undergone STN-DBS surgery at the Department of Neurology, University Medical Center of the Johannes Gutenberg University Mainz. Due to the invasive nature of STN recordings we were not able to record pilot data to compute the effect size of STN-LFP changes related to drawing movements. However, given the very good signal-to-noise ratio of invasive STN LFP recordings with a typical sample size of 10–15[21,24,26,31], we considered a slightly increased sample size appropriate. All participants performed the task with and without the application of high-frequency STN stimulation. We concurrently recorded bilateral STN LFP and applied DBS via externalized electrode extension cables. The diagnosis of PD was based on the Movement Disorder Society Clinical Diagnostic Criteria for PD[53]. The average age was $67.68 \pm 7.25$ years and the pre-operative score on the motor section of the Unified Parkinson's Disease Rating Scale (UPDRS), was $34.16 \pm 14.45$ off and $23.47 \pm 13.18$ on dopaminergic medication. The mean disease duration at the time of the surgery was $10.42 \pm 4.82$ years. Further demographics and disease-related information are presented in Table 1 and individual clinical details are listed in Supplementary Table 1. Lead placement was supported by intra-operative micro-recordings as well as by monitoring the clinical effects and side effects during surgery, and was verified by

postoperative stereotactic computerized topography. We conducted the experiment in the postoperative period two to four days after insertion of the DBS lead and before the implantation of the sub-cutaneous pulse generator, after over-night withdrawal of dopaminergic medication. The implanted leads were either the Medtronic 3389™ DBS leads (Medtronic Neurological Division) with four platinum-iridium cylindrical contacts ($n = 4$), or the directional Abbott 6170™ leads (St. Jude Medical/Abbott) with three segmented contacts on levels 2 and 3 ($n = 15$). All participants gave written informed consent to participate in the study, which was approved by the local ethics committee (State Medical Association of Rhineland-Palatinate) and conducted in accordance with the declaration of Helsinki.

### Graphomotor movement task

**Spiral drawing.** Participants comfortably sat in front of a table and drew Archimedean spirals on a sheet of paper (DIN A4) which was attached to the surface of a digital graphics tablet (Wacom Intuos Pro – Creative Pen Tablet, size L, resolution = 5080 lpi, maximal sample rate 200 Hz, pressure sensitivity = 2048 levels, Wacom Technology Corporation, Vancouver, WA). This setup allowed drawing with a wireless inking pen to provide the natural friction while simultaneously storing digital kinematic time-series data for offline-analyses. Participants drew up to twelve spirals with their dominant hand (18 right-handed patients, one left-handed patient as revealed by self-report), each on predefined templates (five loops, maximal radius = 7.5 cm, spacing = 1.5 cm) as well as blank sheets without any visual guidance (five loops). We recorded both template-guided and free drawings to disentangle motor impairments in highly controlled and less constrained movements: drawing on a template comprises tracking movements that require both speed and accuracy and are often accompanied by one or more corrective sub-movements[54], while drawing freely should shift the speed-accuracy trade-off towards a faster movement execution.

Patients performed the tasks both with and without stimulation, blinded to the condition, in randomized order. To ensure blinding for patients, stimulation was delivered with an intensity below the side-effect threshold. Details about the titration of the stimulation settings are given below. The experimenter (M.B.) instructed the participants to sit with their shoulders in parallel to the lower side of the tablet and draw the spirals at their self-selected drawing speed, starting from the center outwards without touching or crossing the boundaries of the template and without resting the arm or hand on the tablet. Subjects held the pen in the air and started to draw when the experimenter gave a verbal go-command. Right-handed subjects drew the spirals clockwise, while one left-hander drew counter-clockwise.

**Spiral analysis.** We recorded kinematic time-series data using the free software Neuroglyphics (version Oct 1, 2018, http://www.neuroglyphics.org/). We compared the average velocity and spatial accuracy of all trials across all four conditions. We exported positional X- and Y-signals to Matlab (The MathWorks, version R2017a) and applied a fourth-order 10 Hz low-pass Butterworth filter to smooth the signal and reduce artifacts[9]. Then, we calculated the instantaneous tangential velocity with the kinematics toolbox (version V.1.0, http://www.diedrichsenlab.org/toolboxes/toolbox_kinematics.htm). The velocity signals were filtered with a second-order Gaussian Kernel filter[2] and the average velocity of each individual trial was calculated as the mean of the instantaneous velocity signal.

To assess spatial accuracy of motor performance, we first 'unraveled' the original drawings, converting them from the X-Y-representation to the radius-angle-transformation (Supplementary Fig. 1)[55]. Next, we fitted a linear model to each radius-angle-transformation and calculated the root mean squared error (RMSE) and the slope of the regression line. Because the increase of radius is constant for Archimedean spirals after the first revolution, we excluded the first revolution ($-0.5 \pi$ rad to $1.5 \pi$ rad) from the linear model.

The RMSE represents the average deviation from an optimal trace, while the slope represents the average increase of the radius.

### Signal recording and preprocessing

Fig. 1A illustrates the LFP signal recording and processing steps for the experiment. We recorded LFPs from both STN at 2048 Hz using a TMSi-Porti amplifier (TMS International, Netherlands, driver version: 7_2_144). Signals were sampled in a wide bipolar montage to allow for LFP recordings during the stimulation of an interleaved electrode and mitigating the stimulation artifact via common mode rejection[21]. When patients were implanted with the Abbott 6170™, the three directional contacts within one level were joined to form a single contact. Signals were amplified and low-pass filtered at 500 Hz, and the ground electrode was placed on one forearm. The recording was visually inspected for artifacts offline in Spike2 (Cambridge Electronic Design, Cambridge, UK, version 8.10). We identified trials with manually placed markers and concurrently recorded accelerometer data. Then we exported each trial including a six-second interval before and after movement execution. After rejecting trials that could not be unambiguously aligned, a total of 669 trials was included in the LFP-analysis. Further analysis of the data was performed in FieldTrip[56] (version 20220310, https://www.fieldtriptoolbox.org/), as implemented in Matlab. The data were imported to Matlab and high-pass filtered at 4 Hz with a 4th order Butterworth filter. We then down-sampled the data to 200 Hz with an anti-aliasing filter at 100 Hz, demeaned and detrended the data, and applied a discrete Fourier transform filter at 50 Hz and 100 Hz to remove line noise.

### Time-frequency and beta burst analysis

To investigate the neural underpinnings of spiral drawing within the STN, we focused our analysis on beta oscillations and beta bursts. While beta desynchronization is consistently shown before and during force generation tasks[17–20], bursts have been shown to be relevant for movements as well, to be pathologically prolonged in patients with PD, and have been proposed to be a candidate biomarker for adaptive stimulation[24,29,30]. Therefore, we addressed three specific questions: (1) Are beta bursts modulated by drawing movements? (2) Is the drawing velocity affected by the occurrence of beta bursts? (3) Does continuous DBS interfere with bursts during drawing and thus influence movement execution?

To examine the temporal dynamics of broadband beta power within the contralateral STN, we transformed the data to the time-frequency domain with a continuous Morlet wavelet transform (width = 7) for frequencies from 4 Hz to 100 Hz at steps of 1 Hz and 20 ms for the center of the moving window to (Fig. 1B). To account for potential inaccuracies related to the manual synchronization, we excluded ±100 ms of data around the beginning of the trial. Since the drawing duration varied both within and across subjects, we performed a temporal normalization for the time-frequency representation. The power for each time-frequency vector was linearly interpolated such that a five-second rest interval was warped to 50 samples, while the drawing interval was warped to 100 samples with the *timeWarp* function (version 2006) from EEGLab (https://sccn.ucsd.edu/eeglab/index.php)[57]. To assess the movement related beta-desynchronization, we calculated the average warped time-frequency response for each participant within each condition. We tested for differences between two drawing intervals (early and late, 50 samples each) and the rest interval in the beta frequency range by applying cluster based permutation statistics (see below)[58,59].

To evaluate whether movement and stimulation affect broadband beta and gamma power, we conducted a frequency analysis (1 s window length; discrete prolate spheroidal sequences taper; 2 Hz frequency smoothing) during both the rest and drawing intervals, separately. We then calculated the average power of frequencies

ranging from 13 Hz to 30 Hz for the beta band and from 30 Hz to 45 Hz for the gamma band within each trial.

We then investigated the dynamics of beta bursting activity during drawing in the contralateral STN. Figure 1C illustrates the processing steps involved in the discrimination of beta burst activity. We first identified an individual patient's beta frequency (Supplementary Table 1) as the frequency showing the largest desynchronization during the drawing task, or in the case of similar levels between multiple frequencies, the peak frequency during the rest interval[26]. The preprocessed signals were digitally filtered around that individual beta frequency (±2 Hz), rectified, and smoothed with a 200 ms moving average window to calculate the dynamics of the beta envelope amplitude. Beta bursts were defined as consecutive time points at which the amplitude signal exceeded a threshold defined as the 75th percentile of the signal amplitude distribution within the condition-specific (i.e. free or template-guided drawing with and without stimulation) rest intervals[24]. We did not consider bursts shorter than 100 ms to limit the contribution of spontaneous fluctuations in amplitude related to noise. For every burst, we also determined the maximal amplitude. Additionally, we counted the number of bursts within the rest and drawing intervals and calculated the burst rate as the number of bursts divided by the duration.

## High-frequency electrical stimulation

The stimulation electrode was determined in another experiment that was performed the day before where patients had to press a force handle in response to a visual cue[21] ($n = 13$), or by a similar movement task including continuous hand flexion-extension movements ($n = 6$). Here, LFPs were sampled in a monopolar montage. Two wide bipolar channels were computed offline by subtraction. Thus, a dorsal bipolar channel between the most dorsal and second most ventral contact (contact 1 - contact 3) and a ventral bipolar channel between the most ventral and the second most dorsal contact (contact 2 – contact 4) were created. Motivated by evidence linking beta band activity to movement and to the dorsal (motor) region of the STN[60–64], we chose the two electrodes of the channel with the highest movement-related beta modulation as the recording electrodes and the intermediate electrode for stimulation.

We applied DBS in pseudo-monopolar mode using reference pads on the patients' shoulders as anodes via a custom-built device that had been previously validated[21,59]. Frequency (130 Hz) and pulse width (60 µs) were fixed. We performed a test stimulation before the experiment to determine the optimal stimulation currents that provided the best clinical benefit without any side effects. When the threshold for clinical effects was reached, the intensity was noted and, in case of side effects, slightly decreased. A trained movement disorder specialist (G.T., D.H., or D.C.) evaluated this procedure by assessing double-blind upper and lower limb bradykinesia, rigidity and tremor scores (items 3.3, 3.4, 3.5, 3.6, 3.7, 3.8, 3.14, 3.15, 3.16, 3.17, and 3.18) of UPDRS$_{III}$[65] with and without stimulation in randomized order. Because stimulation effects do not appear immediately[23], patients received at least a two minutes wash-in/wash-out period after changing the stimulation condition before the UPDRS$_{III}$ examination or the drawing experiment.

Whilst stimulation was applied, LFPs were continuously recorded via the two contacts neighboring the stimulation contact as described above. Despite common-mode rejection, an artifact was visible. We applied an additional DFT filter at 130 Hz after importing, filtering, and down-sampling. Similar to our previous work[21], we removed segments of data above a threshold of 10 µV (on average ~6% of the signal). The missing samples were then replaced by linear interpolation of the neighboring, non-noisy signals.

## Statistical analysis

If not stated otherwise, all statistical analyses were performed in RStudio, (version 2022.02.1, http://www.rstudio.com/)[66]. The significance threshold was set to $P < 0.05$ (two-sided unless stated otherwise). Illustrations were created with the R ggplot2 package (version 3.4.0, https://ggplot2.tidyverse.org), Fieldtrip, or Matlab. Exact statistics including degrees of freedom, confidence intervals, t-scores, and P-values are provided in Supplementary Tables 3–6.

The effect of stimulation on the UPDRS$_{III}$ was evaluated with paired sample t-tests. Normality was assessed with the Shapiro-Wilk test. Single trial values of velocity, burst duration, and burst amplitude were log-transformed because of their skewed distribution. We applied separate linear mixed-effects models using lme4[67] (version 1.1-32, https://cran.r-project.org/web/packages/lme4/index.html) to investigate the fixed effects of drawing condition (free vs template), stimulation (off vs on), as well as their interaction effects on the average drawing velocity, RMSE, and slope of radius-angle-transform as dependent variables. To account for within-subject variations each subject had its own random intercept.

To assess the movement-related beta desynchronization, we performed cluster-based permutation tests investigating the differences between the drawing interval and the rest interval in each condition[58,59]. An alpha of 0.05 was defined as a cluster-building threshold, and 5000 permutations were performed. Based on previous studies[17–20], we had a clear a-priori hypothesis about the spectral characteristics of STN beta-activity (i.e. we expected a movement-related desynchronization) and therefore considered a single sided cluster alpha threshold of 0.05.

To investigate the effects of stimulation (off vs on), movement (rest vs draw), drawing condition (free vs template), as well as all possible interactions on beta bursts, we applied separate linear mixed-effects models with burst duration, burst amplitude, number of bursts, and burst rate as the dependent variables. The effects on broadband beta and gamma power were also assessed with similar models. While stimulation, movement, drawing condition, and interaction-terms were entered as fixed effects, the subjects' intercepts were entered as random effects. Bonferroni-corrected post-hoc tests were calculated with the *emmeans* function in RStudio.

Subsequently, we investigated the behavioral dynamics in the time-period shortly after the beginning of a burst. To this end, we extracted the velocity signals from 0.25 s before and 0.25 s after the beginning of every burst and calculated the average acceleration within each of these two time windows. For visualization, we averaged these signals' 0.5-s windows within each subject. To test if the occurrence of a burst affects the slope of the velocity dynamics, we performed linear mixed-effects models with the acceleration as the dependent variable; interval (pre vs post), stimulation condition (off vs on), drawing condition (free vs template), as well as all possible interactions as fixed effects; and subjects' intercepts as random effects. To exclude that potential effects are driven by a general reduction in acceleration over time we repeated this analysis with two control conditions. First, we created artificial burst onsets at constant intervals of 50 ms throughout each recording. In this way we were able to test if a general reduction of acceleration over time was present in the investigated time window. As a second control condition, we randomly shuffled the onsets of the original bursts within each trial. This allowed us to test whether there was a general reduction in acceleration when the number of randomly created bursts remained equal.

We also tested if burst amplitude or burst duration related to the extent of velocity alterations: for that, we calculated the difference of acceleration between the post- and pre-interval as the dependent variable in linear mixed-effects models. In addition to stimulation (off vs on) and drawing condition (free vs template), we included beta burst amplitude (or burst duration) and all possible interactions with the beta burst amplitude (or burst duration) as additional fixed factors. Since we were interested in associations within each subject, the slope of the beta burst amplitude (or burst duration) was entered in addition to the intercepts as a random effect.

Finally, we investigated if DBS-related alterations in burst amplitude related to clinical improvement. Here, we calculated the patient specific difference in burst amplitude between off and on stimulation. As our drawing task was performed with the dominant hand, we calculated an individual patient's improvement in corresponding UPDRS$_{III}$ hand-scores (items 3.3, 3.4, 3.5, 3.6, 3.15, 3.16, 3.17) between off and on stimulation. We assessed the spearman rank correlations between clinical improvement and stimulation-related modulations of burst characteristics separately for the rest and drawing intervals as well as for the two drawing conditions. *P*-values were Bonferroni-corrected.

We additionally conducted several control analyses to test the robustness of our results. First, because the pen sometimes left the recording surface or crossed its own trace, we repeated our analyses with such inaccurate trials excluded to assess their impact. Secondly, tremor was the main symptom in seven patients, and one of these patients occasionally presented a slight tremor during drawing (Supplementary Fig. 7). To test if the inclusion of tremor-dominant patients affects our results, we repeated the analysis excluding them. Furthermore, because preceding bursting potentially affects movement speed[26], we additionally tested if including the burst duration, burst amplitude, and burst rate of the baseline interval as fixed factors in the models assessing the association between bursts and immediate reductions in velocity impacts the results. Finally, five participants showed only minor or no clinical improvement of total UPDRS$_{III}$ when receiving DBS during the study. To investigate if DBS still affected drawing performance, we also assessed the effect of stimulation and drawing conditions on the drawing velocity, RMSE, and slope of the radius-angle transform on a single subject level in these patients.

### Reporting summary

Further information on research design is available in the Nature Portfolio Reporting Summary linked to this article.

## Data availability

The data that support the findings of this study are available upon request from the corresponding author. Participant consent allows sharing the data exclusively for scientific purposes, thus we cannot openly deposit the full original dataset online. A minimum example dataset (including scripts) is available on https://github.com/manubange/SpiralBeta.git (https://doi.org/10.5281/zenodo.10795009)[68]. Source data are provided with this paper.

## Code availability

All relevant codes employed in the study can be freely accessed without restriction on https://github.com/manubange/SpiralBeta.git.

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

## Acknowledgements

We thank the participants for their time and effort. We thank Peter Brown for providing the stimulation device, conceptual support of the project and critical discussion of the results. We thank Kathleen Claussen for

proofreading the manuscript. The authors would like to acknowledge the support of FTN/TransMed, the German Research Foundation (DFG; CRC-TR-128, S.G.), and the Boehringer Ingelheim Fonds (BIF-03, S.G.). D.M.H. is supported by a postdoctoral grant from the Independent Research Fund Denmark (0168-00014B, D.M.H.) and has received the Boehringer-Ingelheim-Preis. G.T. received funding from the Swiss National Science Foundation (project number: PZ00P3_202166, G.T.). H.T. and A. P. are supported by the Medical Research Council (MRC), UK (MC_UU_00003/2, H.T. and A. P.).

## Author contributions

Conception and design: M.B., G.G.-E., S.G. Acquisition, analysis, and interpretation of data: M.B., G.G.-E., D.M.H., G.T., D.C., M.G., A.P., H.T., S.G. First draft of manuscript: M.B. Revision of manuscript: G.G.-E., D.M.H., G.T., M.G., D.C., A.P., S.L.K., H.J.L, H.T., S.G.

## Funding

## Competing interests

S.G. received financial support from Abbott not related to this work. G.T. received financial support from Boston Scientific and Medtronic not related to the present work. Research agreement with RuneLabs is not related to the present work. The remaining authors declare no competing interests.
