## [Peer Review File · Nature Communications]

Subthalamic stimulation modulates context-dependent effects of beta bursts during fine motor controlREVIEWER COMMENTS

Reviewer #1 (Remarks to the Author):

This was quite a brilliant paper by Tan and colleagues. They cleverly used recordings of immediately externalized DBS leads after the operation. Though this technique can introduce artifact, this group has convincingly shown that the data was reliable across all but two subjects who developed fatigue.

The diverse and complex role of STN beta bursts applied to graphomes and drawing movements was really interesting and well controlled. The authors also nicely pointed out where previous literature provided clarity and why their experiments moved these questions to the next level.

The use of specific task constraints was appropriate and the figures clear.

Bursts were detected during drawing and their occurrence was accompanied by velocity reductions. Thus they convinced me that beta bursts change and play a role in movement execution and impairment in a context-dependent framework.

When they applied DBS the amplitude of bursts reduced in the rest interval and during task execution.

The one thing that could be added to this nice work would be the same tasks on and off medication however I do not think the authors should be held to this standard as externalized DBS is uncomfortable for patients and they got 17/19 patients data. Perhaps in a followup they could do as a separate experiment however I think this paper stands on its own.

Reviewer #2 (Remarks to the Author):

The Authors describe how subthalamic stimulation modulates context-dependent effects of beta bursts during fine motor control in Parkinson's Disease. Main results are that deep brain stimulation decouples beta bursts from acceleration reductions in addition to a general increase in drawing velocity and improvements of clinical function.

The experimental procedure is well described, and enough details are provided to reproduce the experiments. Methods are sound and discussion is interesting and well written. I have only few suggestions the Authors may want to consider:

- 1) METHODS. The Authors may want to describe if a power analysis was performed before starting the study. It is not clear if the patients were included step-by-step to finally reach a given result. Please describe better in the method section.
- 2) METHODS/RESULTS: (page 5). Reduced beta power during drawing. The Authors had a clear a-priori hypothesis about the spectral characteristics of STN beta-activity (i.e. they expected a movement-related desynchronization) and therefore they considered a single sided cluster alpha threshold of 0.05. I agree that this is correct but I suggest to report the exact p value so the reader can judge the robustness of the result.
- 3) FIGURES: Figure 6 (and supplementary figure 5). Please provide how the outlier was identified. Visually? By statistical methodology?
- 4) Abstract: spiral drawing,. Please correct!

Reviewer #3 (Remarks to the Author):

The authors investigated the correlation between beta bursts measured in the subthalamic nucleus and the parameters of two complex hand movements in Parkinson's disease, without and with deep brain stimulation. The patients drew a spiral with the dominant hand, on a template, or freehand. The results are new, in line with the previous literature, promising, and carry important information for developing closed-loop deep brain stimulation.

The results:

Beta burst duration decreased during movement compared to rest and was not affected by DBS treatment.

The burst amplitude decreased during movement, especially under stimulation and when applying a template.

The burst rate was lower when drawing a spiral.

Smaller burst amplitude resulted in higher voluntary movement speed during stimulation.

Greater burst amplitude reduction during DBS in the resting state predicted higher speed drawing.

The number of subjects, the method used, data processing, and statistics are appropriate. The analysis supports the conclusions drawn. The interpretation could be more straightforward; it is difficult to read the text and find the main results in the text among the technical terms. The protocol figure needs

improvement, but the other tables and figures are excellent and informative. The description of the methods is sufficiently detailed.

Comments:

Throughout the manuscript, the change in the beta burst parameters mainly affected the self-paced movements, while the drawing following the template was affected less. In connection with this, the authors concluded that the determining role of bursts in movement performance is context-dependent. The latter statement is, of course, correct but too general. Self-paced movement is primarily regulated by the SMA, which is most affected by dopamine deficiency. In cued movements, the role of the premotor cortex is essential, which is less affected by dopamine deficiency and the associated network activity changes. The latter fact (e.g., application of light signals to start a movement) is also used in the rehabilitation of patients. A more detailed discussion of this issue is recommended in the conclusion highlighting that the stimulation is applied in parallel but different sub-networks, which should be considered when analysing different movement tasks.

1. Abstract: could be more specific and precise, especially concerning the patient group, methods, and results.
2. The introduction and the research question are appropriate.
3. In formulating the results, it would be beneficial to highlight the main results in the text and reduce the statistical expressions and descriptions to make them more readable. The latter is already listed in the table.
4. The discussion is adequate except for the previous comment; many results are clearly stated only here - the results section is difficult to read.
5. The description of the methods is excellent. It is outstanding that the authors selected the movement-reactive beta band for each patient instead of using a standard band. (Ref: Muthuraman et al. Source analysis of beta-synchronization and cortico-muscular coherence after movement termination based on high resolution electroencephalography. Plos One, 2012;7(3):e33928. Analysis on the cortical level.)
6. Figure 1 . To label the illustration (what kind of signal, axis titles, units)

In summary, this is a precisely planned and executed work. The structure of the text could be more logical, highlighting the main results, and its wording could be more understandable.

After a revision, I recommend publishing the manuscript in the journal.

Reviewer #4 (Remarks to the Author):

Thank you for inviting me to review the paper “Subthalamic stimulation modulates context-dependent effects of beta bursts during fine motor control” by Bange et al. In their study, the authors investigate the pattern of subthalamic beta bursting during two continuous movement conditions (free drawing and drawing following a template) both with and without DBS in 19 PD patients. They show that 1 - movements are faster in the free drawing condition as well as during DBS, 2 - beta bursting is reduced during movement and during DBS and 3 – the link between beta bursting and reduced acceleration is present solely in the free drawing condition without DBS. The paper is well written, the figures are a good summary of the results, and the applied methods are to my understanding correctly chosen.

Taken together, this is a sound paper with valid results, but I fear that it may not provide significant novelty with respect to previous publications or increase the mechanistic insight to a level that would justify a publication in Nature communications.

Behaviorally, it is well known that DBS increases movement speed (among others, also shown in Kehnemouyi et al., 2021 or Kehnemouyi et al., 2023). On a neurophysiological level, the authors point out themselves, that Lofredi et al. (2019) already showed that subthalamic beta bursts occur during sustained movements and that they are associated with a decrease in movement speed – which is one of the main findings of the present study. With respect to the DBS effect on beta bursts during movement, Kehnemouyi et al. (2021) showed that the link between beta bursting during movement and bradykinesia is maintained during DBS, which contradicts the results of the present study and should at least be mentioned. Thus, the main findings of this study have already been investigated in previous studies and although a replication of findings is encouraging and should always be published, I fear that the specific additional knowledge gained by this study might be better suited for a journal with a more specialized audience than Nature Communications.

Beyond that, I have relatively few further comments as the authors have provided a well-designed and well-analyzed paper. Please find some comments that might improve the paper even further:

1. It was unclear to me, why the authors decided to include the “template-based” condition, as they did not analyze this condition further with respect to accuracy or error-rates – although this is addressed in the discussion as rationale behind including this condition. I would either considering removing the template condition or include additional analyses with respect to neurophysiological read-outs of behavioral metrics such as accuracy.

2. I would welcome it, if the authors would show the absolute durations of beta bursts per condition. Given that they defined beta bursts as 75th percentile above the amplitude distribution of the rest

period within the same condition, it is likely that the detected bursts during movement in the ON-DBS condition are much shorter than in the OFF-DBS condition – which could explain the lack of a significant link between deceleration and beta bursting in this condition.

3. In the discussion, the authors mention that previous bursting could explain movement slowness. If they think that, they should include this feature in their analysis.

4. Did any of the patients have tremor during drawing? In 7 patients, the main symptom was indicated as being tremor, thus, the authors should investigate how tremor-dominant patient affected their results – and whether they remain robust after removing tremor patients or how performance with DBS changed in these patients.

5. In the same regard, I noticed that 3 patients did not have any DBS effect and 1 patient a very minor DBS-effect. This is a relevant number in this (relatively) small cohort, and the authors should at least show that these patients still had a task-specific behavioral DBS-effect that may not have been captured in the clinical domain (which might be linked to the very short wash-in period of 2 minutes stated by the authors).

6. The authors state that their results will be helpful for clinical deep brain stimulation, although they show that DBS annuls the link between deceleration and beta bursts. How should this finding then be helpful?

Response to the reviewers' comments

We thank the reviewers very much for their support and for the very constructive feedback on our manuscript NCOMMS-23-32552 titled "Subthalamic stimulation modulates context-dependent effects of beta bursts during fine motor control". We have addressed all their concerns and amended our work. We provide our detailed **responses** below, with the newly added text underlined.

We significantly expanded our analyses, including spatial and trial accuracy analyses, adding a further layer of novelty to our manuscript. In the new version of the work, we rephrased and further developed larger parts of the results and discussion sections to extend the implications of this work for our understanding of fine motor control. We are confident that the revisions considerably improve the manuscript and we hope that it is ready for publication.

Reviewer #1 (Remarks to the Author):

This was quite a brilliant paper by Tan and colleagues. They cleverly used recordings of immediately externalized DBS leads after the operation. Though this technique can introduced artifact, this group has convincingly shown that the data was reliable across all but two subjects who developed fatigue. The diverse and complex role of STN beta bursts applied to graphemes and drawing movements was really interesting and well controlled. The authors also nicely pointed out where previous literature provided clarity and why their experiments moved these questions to the next level.

The use of specific task constraints was appropriate and the figures clear.

Bursts were detected during drawing and their occurrence was accompanied by velocity reductions. Thus they convinced me that beta bursts change and play a role in movement execution and impairment in a context-dependent framework.

When they applied DBS the amplitude of bursts reduced in the rest interval and during task execution.

The one thing that could be added to this nice work would be the same tasks on and off medication however I do not think the authors should be held to this standard as externalized DBS is uncomfortable for patients and they got 17/19 patients data. Perhaps in a followup they could do as a separate experiment however I think this paper stands on its own.

Response:

We thank the reviewer for the appreciation and very positive feedback and their interest in the role of beta bursts on motor control and especially during drawing movements with different task constraints. As the reviewer mentioned, performing experimental procedures is demanding for patients with externalized DBS electrodes. To reduce the discomfort for the patients and exclude range effects and the interaction of the microlesional effect after implantation with the

medication state we did not repeat our study in the medication ON condition. However, we absolutely agree that the effects of medication on the relationship between beta bursts and velocity dynamics and the impact of DBS constitute a relevant question, given that levodopa intake also affects beta burst characteristics and improves bradykinesia. In this context, previous work showed that medication does not influence drawing velocity in patients with PD (Danna et al., 2018). Unfortunately, in this study patients either drew only one single spiral per condition or the drawing velocity was averaged within each participant, reducing the implications of the reported analyses for our work. In the revised version of the work we added an extensive passage on the necessity to investigate the effects of medication on the relationship between beta bursts and velocity dynamics for different types of movement and DBS in the discussion section:

Revisions in the discussion section (page 17):

In this context, DBS increased the acceleration around beta bursts, modifying their immediate ramifications on velocity dynamics during drawing in PD. The modulation of such relationship under stimulation corroborates the notion that DBS-related modulations of beta bursts might underlie the improvement of the progressive deterioration of velocity during ongoing movement in chronically implanted patients.⁴⁷ However, the acute intake of levodopa also reduces duration, amplitude, and rate of subthalamic beta bursts,⁴⁶ and it remains to be elucidated how the effect of DBS might be modulated by levodopa.

Reviewer #2 (Remarks to the Author):

The Authors describe how subthalamic stimulation modulates context-dependent effects of beta bursts during fine motor control in Parkinson's Disease. Main results are that deep brain stimulation decouples beta bursts from acceleration reductions in addition to a general increase in drawing velocity and improvements of clinical function.

The experimental procedure is well described, and enough details are provided to reproduce the experiments. Methods are sound and discussion is interesting and well written. I have only few suggestions the Authors may want to consider:

1) METHODS. The Authors may want to describe if a power analysis was performed before starting the study. It is not clear if the patients were included step-by-step to finally reach a given result. Please describe better in the method section.

Response:

We thank the reviewer for the positive feedback and agree on the suggestion that the sample size estimation should be included in the methods section. Due to the invasive nature of our recordings we were not able to record pilot data for computing the effect size of STN-LFP changes related to drawing movements. However, given the very good signal-to-noise ratio of invasive STN LFP recordings with a typical sample size of 10-15 in similar work (see refs in the main text 21, 24, 26 and 31 in the article) we considered a slightly increased sample size appropriate. With the reviewers suggestion we have nevertheless added the following paragraph in the methods section:

Revisions in the results section (page 20):

We investigated the dynamics of beta bursts during a spiral-drawing task in 19 participants with PD who had undergone STN-DBS surgery at the Department of Neurology, University Medical Center of the Johannes Gutenberg University Mainz. Due to the invasive nature of STN recordings we were not able to record pilot data to compute the effect size of STN-LFP changes related to drawing movements. However, given the very good signal-to-noise ratio of invasive STN LFP recordings with a typical sample size of 10-15,^{21,24,26,31} we considered a slightly increased sample size appropriate. All participants performed the task with and without the application of high-frequency STN stimulation.

2) METHODS/RESULTS: (page 5). Reduced beta power during drawing. The Authors had a clear a-priori hypothesis about the spectral characteristics of STN beta-activity (i.e. they expected a movement-related desynchronization) and therefore they considered a single sided cluster alpha threshold of 0.05. I agree that this is correct but I suggest to report the exact p value so the reader can judge the robustness of the result.

Response:

We have added the Supplementary Table 2 to report the cluster statistics including the exact P-values:

Supplementary Table 2. Exact cluster-statistics for the movement related desynchronization in in the beta band in both stimulation and drawing conditions.

drawing condition	stimulation condition	interval	cluster-number	cluster statistics	SD	CI range	P
free	Off	early	1	-232.053	<0.001	0.001	0.001
free	Off	early	2	-174.918	0.001	0.002	0.006
free	Off	late	1	-321.926	0.001	0.002	0.004
free	Off	late	2	-296.958	0.001	0.002	0.005
free	Off	late	3	-140.574	0.002	0.004	0.018
free	Off	late	4	-129.026	0.002	0.004	0.021
free	On	early	1	-264.435	0.001	0.003	0.010
free	On	early	2	-251.884	0.002	0.003	0.013
free	On	late	1	-184.277	0.002	0.005	0.027
template	Off	early	1	-188.326	0.000	0.001	0.000
template	Off	early	2	-80.760	0.003	0.006	0.043
template	Off	late	1	-72.885	0.002	0.005	0.030
template	Off	late	2	-69.128	0.003	0.005	0.035
template	On	early	1	-435.262	0.000	0.001	0.001
template	On	early	2	-268.146	0.001	0.003	0.009
template	On	early	3	-162.573	0.003	0.006	0.048
template	On	late	1	-210.178	0.002	0.004	0.021

Only significant clusters are reported. Cluster-statistic: cluster-level test statistic (the sum of the T-values in this cluster); SD: standard deviation of the probability values; CI range: range of the confidence interval of the probability values. P: P-value

3) FIGURES: Figure 6 (and supplementary figure 5). Please provide how the outlier was identified. Visually? By statistical methodology?

Response:

We agree with the reviewer that the identification of the outlier should be mentioned in the manuscript. We determined the outlier as ± 3 standard deviations from the mean, which was originally exceeded by one subject in the free drawing condition. However, during the revision we corrected a small inaccuracy in the script determining the beta-threshold, which prompted us to repeat all analyses with the revised version. While our main finding that single beta bursts parallel acceleration reductions exclusively during free drawing is not affected by this, no

outlier was identified in the clinical correlations anymore. We thus excluded supplementary Figure 6 from the manuscript.

4) Abstract: spiral drawing,. Please correct!

Response:

We have corrected this typo.

Reviewer #3 (Remarks to the Author):

The authors investigated the correlation between beta bursts measured in the subthalamic nucleus and the parameters of two complex hand movements in Parkinson's disease, without and with deep brain stimulation. The patients drew a spiral with the dominant hand, on a template, or freehand. The results are new, in line with the previous literature, promising, and carry important information for developing closed-loop deep brain stimulation.

The results:

Beta burst duration decreased during movement compared to rest and was not affected by DBS treatment.

The burst amplitude decreased during movement, especially under stimulation and when applying a template.

The burst rate was lower when drawing a spiral.

Smaller burst amplitude resulted in higher voluntary movement speed during stimulation.

Greater burst amplitude reduction during DBS in the resting state predicted higher speed drawing.

The number of subjects, the method used, data processing, and statistics are appropriate. The analysis supports the conclusions drawn. The interpretation could be more straightforward; it is difficult to read the text and find the main results in the text among the technical terms. The protocol figure needs improvement, but the other tables and figures are excellent and informative. The description of the methods is sufficiently detailed.

Response:

We thank the reviewer for the support, appreciation and positive feedback that helped us to further improve the manuscript. We have extended the interpretation of our findings by including a discussion about the differences between internally guided and externally cued movements and the different networks that are involved (please see our response to the first general comment). Furthermore, we revised the results section to better highlight the main results (see our response to comment 3) and included more details in the protocol figure (see our response to comment 6).

Comments:

Throughout the manuscript, the change in the beta burst parameters mainly affected the self-paced movements, while the drawing following the template was affected less. In connection with this, the authors concluded that the determining role of bursts in movement performance is context-dependent. The latter statement is, of course, correct but too general. Self-paced movement is primarily regulated by the SMA, which is most affected by dopamine deficiency. In cued

movements, the role of the premotor cortex is essential, which is less affected by dopamine deficiency and the associated network activity changes. The latter fact (e.g., application of light signals to start a movement) is also used in the rehabilitation of patients. A more detailed discussion of this issue is recommended in the conclusion highlighting that the stimulation is applied in parallel but different sub-networks, which should be considered when analysing different movement tasks.

Response:

We thank the reviewer for the suggestion to elaborate on the differences between uncued/internally guided and externally cued movements, involved networks, and how these distinct circuits are modulated by DBS in more detail. The suggestion to further develop on the recruitment of supplementary motor circuits for template-guided drawing is also in our view a very interesting aspect, which we now extensively address in the discussion section. Because of the strong dependency of internally guided movements on basal-ganglia-SMA networks, it is reasonable that these movements are particularly affected by STN-DBS.

Revisions in the discussion section (page 15):

Additionally, free drawing represents an internally guided movement, while presenting a template provides an external visual cue. Importantly, people with PD have particular difficulties with internally guided movements, and the presentation of visual or auditory cues can help patients to overcome movement impairments.^{40, 41} Externally guided movements are thought to preferentially involve cerebellar, premotor, and sensory circuits, while the basal ganglia and supplementary motor area (SMA), which are heavily affected by dopamine deficiency, play a more important role for internally guided movements.⁴²⁻⁴⁵ The shift towards potentially less impaired motor circuits has been hypothesized to underlie the positive effects of external cueing strategies for addressing gait impairments in PD.³⁴ Likewise, presenting a template might engage supplementary pathways such as the cerebello-thalamo-cortical circuit in parallel, thereby influencing or surpassing the role of basal-ganglia beta oscillations. However, when comparing neural activation during tracing and free drawing movements with functional magnetic resonance imaging, the predicted shift in activation patterns could be shown in the pre-SMA but not in the cerebellum.⁴⁴ Thus, more research is needed to investigate temporo-spatial differences in electrophysiological activities between free and guided drawing movements.

Revisions in the discussion section (page 17):

Patients also increased the average drawing velocity during template-guided drawing when receiving DBS, although bursts did not present the same immediate effects on the velocity dynamics as during free drawing. Considering that template-guided drawing is an externally guided movement that likely engages cerebello-thalamo-cortical networks,⁴²⁻⁴⁴ DBS-related modulations of communication between STN and (pre-)SMA^{48, 49} can be expected to have a more pronounced impact on free, internally guided drawing movements that predominantly rely on basal ganglia-cortical networks. While our results demonstrate that DBS increased

drawing velocity and decreased burst amplitude in both conditions, at least one additional mechanisms must play a role. Recent evidence suggests that average power and burst-characteristics can provide complementary information.⁵⁰ Nevertheless, we did not find any associations between the extent of movement-related desynchronization in the broadband beta range and the drawing velocity. Further investigation is necessary to determine whether the modulation of separate yet inter-connected pathways engaged in template-guided drawing contribute to a stimulation-related increase in drawing speed.

1. Abstract: could be more specific and precise, especially concerning the patient group, methods, and results.

Response:

Many thanks for this suggestion. While we abstained from including too technical details about the methods due to the limit of 200 words according to the submission guidelines, we agree that some additional information would be beneficial in the abstract. Thus, we revised the abstract to provide relevant information about the patient group, methods, and results:

Revisions in the abstract (page 1):

Increasing evidence suggests a considerable role of pre-movement beta bursts for motor control and its impairment in Parkinson's disease. However, whether beta bursts occur during precise and prolonged movements and if they affect fine motor control remains unclear. To investigate the role of within-movement beta bursts for fine motor control, we here combine invasive electrophysiological recordings and clinical deep brain stimulation in the subthalamic nucleus in 19 patients with Parkinson's disease performing a context-varying task that comprised template-guided and free spiral drawing. We determined beta bursts in narrow frequency bands around patient-specific peaks and assessed burst amplitude, duration, and their immediate impact on drawing speed. We reveal that beta bursts occur during the execution of drawing movements with reduced duration and amplitude in comparison to rest. Exclusively when drawing freely, they parallel reductions in acceleration. Deep brain stimulation increases the acceleration around beta bursts in addition to a general increase in drawing velocity and improvements of clinical function. These results provide evidence for a diverse and task-specific role of subthalamic beta bursts for fine motor control in Parkinson's disease; suggesting that pathological beta bursts act in a context dependent manner, which can be targeted by clinical deep brain stimulation.

2. The introduction and the research question are appropriate.

Response:

We thank the reviewer for their positive feedback

3. In formulating the results, it would be beneficial to highlight the main results in the text and reduce the statistical expressions and descriptions to make them more readable. The latter is already listed in the table.

Response:

We have removed the t-scores from the statistical tests as well as the individual test results from post-hoc analyses whenever possible to highlight the main results and improve readability. The extended test-statistics are now provided Supplementary Tables 3 and 4. To further highlight the main results we added brief summaries of the results below several paragraphs in the results section:

Revisions in results section (page 7):

These results show that patients generally drew slower when a template was provided, which could be attributed to an increase of accuracy constraints. Applying DBS in the STN generally sped up drawing movements.

Revisions in results section (page 12):

Together, these results demonstrate that beta bursts parallel reductions in acceleration when drawing freely. Additionally, the implementation of DBS specifically modifies movement acceleration around beta bursts in this drawing context.

4. The discussion is adequate except for the previous comment; many results are clearly stated only here - the results section is difficult to read.

Response:

We thank the reviewer for their acknowledgement. Please see our response to comment 3 regarding the results section.

5. The description of the methods is excellent. It is outstanding that the authors selected the movement-reactive beta band for each patient instead of using a standard band. (Ref: Muthuraman et al. Source analysis of beta-synchronization and cortico-muscular coherence after movement termination based on high resolution electroencephalography. Plos One, 2012;7(3):e33928. Analysis on the cortical level.)

Response:

We thank the reviewer for their positive feedback about the presentation of the methods. We have included the suggested reference that provides additional impetus to focus on individual, rather than broad standard bands:

To investigate the dynamics of beta bursting activity during drawing in the contralateral STN, we performed a similar analysis to previous work.^{24, 26, 59}

6. Figure 1. To label the illustration (what kind of signal, axis titles, units)

Response:

As suggested by the reviewer, we have now better labeled the illustrations in Figure 1. Please see below the revised figure.

Figure 1. Experimental setup and analytical steps. (A) Patients drew spirals with their dominant hand on a digital tablet while we recorded local field potentials (LFP) from the bilateral subthalamic nuclei in a ‘wide’ bipolar montage. This allowed us to deliver deep brain stimulation simultaneously from the interleaved contact. The task was performed with and without stimulation, in randomized order. An example of the tangential velocity and the preprocessed LFP (high- and lowpass-filter at 4 Hz and 100 Hz; down-sampling; DFT-filter; demeaning and detrending) are presented as a function of time (right). After offline preprocessing the LFP signals were analyzed in two different steps. **(B)** LFP-signals were transformed to the time-frequency representation (TFR) from 4 Hz to 100 Hz with a frequency resolution of 1 Hz and 20 ms for the center of the moving window.²¹ **(C)** The LFP signal was filtered around individually determined beta frequencies (Supplementary Table 1), rectified, and smoothed to obtain the envelope of the beta activity. For each condition, a threshold was then set at the 75th percentile of the beta amplitude of the corresponding rest interval. The onset of a burst was defined as when the rectified signal crossed the threshold amplitude while the end of the burst was defined as when the amplitude fell below the threshold. All bursts with a duration longer than 100 ms were considered.

In summary, this is a precisely planned and executed work. The structure of the text could be more logical, highlighting the main results, and its wording could be more understandable.

After a revision, I recommend publishing the manuscript in the journal.

Response:

We thank the reviewer for their comments and included the suggested points. We discuss the task-dependent differential recruitment of brain regions, revised the abstract and results section, and added labels to Figure 1. We are convinced that this improved the comprehension of the implications of our findings.

Reviewer #4 (Remarks to the Author):

Thank you for inviting me to review the paper "Subthalamic stimulation modulates context-dependent effects of beta bursts during fine motor control" by Bange et al. In their study, the authors investigate the pattern of subthalamic beta bursting during two continuous movement conditions (free drawing and drawing following a template) both with and without DBS in 19 PD patients. They show that 1 - movements are faster in the free drawing condition as well as during DBS, 2 - beta bursting is reduced during movement and during DBS and 3 - the link between beta bursting and reduced acceleration is present solely in the free drawing condition without DBS. The paper is well written, the figures are a good summary of the results, and the applied methods are to my understanding correctly chosen.

Taken together, this is a sound paper with valid results, but I fear that it may not provide significant novelty with respect to previous publications or increase the mechanistic insight to a level that would justify a publication in Nature communications.

Behaviorally, it is well known that DBS increases movement speed (among others, also shown in Kehnemouyi et al., 2021 or Kehnemouyi et al., 2023). On a neurophysiological level, the authors point out themselves, that Lofredi et al. (2019) already showed that subthalamic beta bursts occur during sustained movements and that they are associated with a decrease in movement speed - which is one of the main findings of the present study. With respect to the DBS effect on beta bursts during movement, Kehnemouyi et al. (2021) showed that the link between beta bursting during movement and bradykinesia is maintained during DBS, which contradicts the results of the present study and should at least be mentioned. Thus, the main findings of this study have already been investigated in previous studies and although a replication of findings is encouraging and should always be published, I fear that the specific additional knowledge gained by this study might be better suited for a journal with a more specialized audience than Nature Communications.

Beyond that, I have relatively few further comments as the authors have provided a well-designed and well-analyzed paper. Please find some comments that might improve the paper even further:

Response:

We thank the reviewer for the very kind appreciation and very helpful feedback. It helped us a lot to further improve this manuscript and we hope that with additional analyses and elaborations we could gain your ultimate appreciation on this work. We greatly agree on the relevance of the Kehnemouyi et al. (2021) and Kehnemouyi et al. (2023) manuscripts on beta-bursts and the DBS-dependent modulation of movement speed, and kindly apologize for not discussing this highly relevant work in the previous version of the manuscript. These articles helped us to bring an additional level of interpretation to our results, further elaborating on the putative mechanisms underlying the differential effects of beta bursts on movement speed under different task conditions (i.e. drawing with and without

visual cues). Especially together with the newly included analyses on the effects of DBS on spatial accuracy (see point 1 below), we are convinced that our work provides novel findings and is interesting for a broad readership.

We agree that previous work has established the relationship between beta bursts and bradykinesia. This, in fact, motivated us to design the study in a way that allows us to provide novel and complementary insights. Expanding on previous discoveries focused on repetitive movements (for example pronation-supination or flexion-extension of the wrist) we aimed to investigate activities that impose greater demands on fine motor control, closely mimicking real-world scenarios, and thus promoting the applicability and relevance in practical situations. In this context, the task of drawing poses increased demands on motor system function such as the precise coordination of multiple effectors or the integration of sensorimotor feedback, yet is independent from linguistic skills (Danna et al., 2011, Danna et al., 2018). Furthermore, we incorporated two conditions that posed different demands on accuracy, as indicated by a hypothesized speed accuracy trade-off (see also comment 1). In particular, the two conditions also differed in terms of how visual feedback is used to guide the movement planning and execution. In contrast, most previous studies investigating the role of beta bursts in movements used either brief ballistic movements (Torrecillos et al., 2018, Torrecillos et al., 2023), or self-paced free, stereotyped movements (Lofredi et al., 2018, Lofredi et al., 2019, Kehnemouyi et al., 2021). While we implicitly assumed that providing a template would shift the speed-accuracy trade-off, we further conducted an analysis to verify whether this assumption holds true.

Importantly, we believe that our results do not contradict the findings in Kehnemouyi et al. (2021), work that investigated the relationship between average burst duration and average speed when modulating the stimulation intensity. We were instead interested in how individual beta bursts immediately affect drawing velocity and how this is modulated by optimal stimulation. Investigating the immediate effects of single bursts on the velocity dynamics instead of average velocity provides an increased temporal precision and is thus also related to the sequence effect (the progressive decrement of speed during ongoing movement), as studied in Kehnemouyi et al. (2023). We thus demonstrate that a direct link between single beta bursts and immediate reductions in acceleration is only present during free drawing movements but not in the condition when visual feedback is used to guide the movements. We think this is a crucial finding for evaluating the generalizability of previous observations.

As mentioned for reviewer #2 (comment 3), we reanalyzed our data with corrected beta thresholds. This did not affect our main finding showing that beta bursts are accompanied by immediate reductions in acceleration exclusively during free drawing. In the newly performed analysis, however, the three-way interaction is present only as a trend ($P = 0.075$). Thus, the data do not provide absolute evidence that DBS completely disrupts the interrelation between single bursts and immediate reductions in acceleration. Instead, continuous DBS might increase the acceleration around beta bursts during free drawing in line with the proposed role of DBS-related modulations of beta bursts underlying the improvement of the

sequence effect (Kehnemouyi et al., 2023). We have now updated the introduction, results, figure 5, and included these relevant aspects in the discussion section:

Revisions in the introduction (page 4):

Altogether, we aimed to investigate the functional relationship between the execution of a spiral drawing task and STN beta activities in patients with PD, and how this is affected by DBS. Because the presentation of visual or auditory stimuli modulates motor programs and these cues can alleviate parkinsonian symptoms,^{33,34} we designed a task where the participants were asked to draw spirals under two distinct conditions: drawing freely, which represents an internally guided movement, and drawing with a template that provides additional external visual cues. The two tasks require varying levels of accuracy and affect the contributions of sensory-visual feedback.

Revisions in the results section (page 11):

We found significant main effects of the time interval ($P < 0.001$) and the stimulation condition ($P = 0.001$), but not the drawing condition ($P = 0.415$). We further found interaction effects for stimulation condition * drawing condition ($P = 0.007$) and time interval * drawing condition ($P = 0.019$). Following the two two-way interactions, the post hoc tests showed that the effects of time interval and stimulation condition were only present during free drawing ($P = 0.002$ and $P = 0.008$, respectively).

Revisions in Figure 5 (page 42):

Figure 5. Deep brain stimulation alleviates burst related reductions of acceleration during free drawing. (A) Top: For visualization, velocity signals within windows of 250 ms before and after the occurrence of a burst were normalized by the instantaneous velocity during burst onset and averaged within each subject. Average velocity signals and the standard error of the mean (shaded area) for all drawing and stimulation conditions (blue = free drawing without stimulation; red = free drawing with stimulation; green = template-guided drawing without stimulation; orange = template-guided drawing with stimulation) are shown. We calculated the average acceleration within the pre and post intervals for each burst. Linear mixed effect models showed main effects of time interval and stimulation condition and interaction effects for stimulation condition * drawing condition, and drawing condition * time interval). Post hoc tests demonstrated that only bursts occurring during free drawing are accompanied by a reduction in acceleration (as can be seen as a flattening of the velocity signal after the occurrence of a burst). When applying DBS, the acceleration around beta bursts was generally increased during free drawing, suggesting that stimulation modifies their immediate impact on velocity dynamics. When drawing with a template, acceleration was not affected by the occurrence of a burst. Bottom: modelled beta amplitude dynamics demonstrating the definition of the burst onset as the time point when the amplitude crosses the 75th percentile threshold. **(B)** Velocity dynamics for the two control conditions showing that the velocity signal does not change its slope after the occurrence of linearly or randomly defined onsets (top: burst onsets defined in 50 ms steps across every trial; bottom: randomly assigned timestamps). Linear mixed effect models did not show any significant effects, highlighting that the average acceleration after the defined onsets is not significantly different from before.

Revisions in the discussion section (pages 15-16)

Accordingly, patients reduced the drawing velocity in comparison to free drawing, potentially to integrate visual feedback and exert active control to flexibly adjust the pen-movements. Supporting the presence of such a speed-accuracy trade-off, our results demonstrate that the presentation of the template was accompanied by reductions in deviations from an optimal trace and increases in the spiral's diameter. Importantly, RMSE and slope were directly associated with the drawing

velocity. Consequently, the general reduction in movement velocity might mask the effects of beta bursts because pathological decrements are relatively small in relation to the average velocity and thus lose their relevance in highly controlled and slow movements. Furthermore, beta activity might play a different or additional role during feedback-guided in comparison to free movements. For example, it was shown that the integration of visual feedback modulates both cortical and subcortical beta oscillations, indicating that the regulation of beta activity also reflects neural processes related to gating feedback and evaluating movement deviations.³⁹⁻⁴¹ Consequently, an increased reliance on visual feedback posed by the template might modulate the role of beta bursts and thus influence the impact of single bursts on the velocity dynamics. Additionally, free drawing represents an internally guided movement, while presenting a template provides an external visual cue. Importantly, people with PD have particular difficulties with internally guided movements, and the presentation of visual or auditory cues can help patients to overcome movement impairments.^{33, 34} Externally guided movements are thought to preferentially involve cerebellar, premotor, and sensory circuits, while the basal ganglia and supplementary motor area (SMA), which are heavily affected by dopamine deficiency, play a more important role for internally guided movements.⁴²⁻⁴⁵ The shift towards potentially less impaired motor circuits has been hypothesized to underlie the positive effects of external cueing strategies for addressing gait impairments in PD.³⁴ Likewise, presenting a template might engage supplementary pathways such as the cerebello-thalamo-cortical circuit in parallel, thereby influencing or surpassing the role of basal-ganglia beta oscillations. However, when comparing neural activation during tracing and free drawing movements with functional magnetic resonance imaging, the predicted shift in activation patterns could be shown in the pre-SMA but not in the cerebellum.⁴⁴ Thus, more research is needed to investigate temporo-spatial differences in electrophysiological activities between free and guided drawing movements.

Revisions in the discussion section (page 16-17)

The modulatory effect of continuous subthalamic stimulation on burst duration is less clear. In line with our results, Tinkhauser et al.²⁴ showed that conventional DBS does not alter burst duration during rest. Using a different approach for burst detection and a rigorous withdrawal of dopaminergic medication, Kehnemouyi et al.³² showed that DBS reduces burst duration during repetitive flexion-extension movements of the wrist in chronically stimulated PD patients. Thus, it is possible that residual effects of medication or a surgery-related temporal lesioning effect³⁵ masks the modulation of beta burst duration. Kehnemouyi et al.³² also demonstrated that the DBS-related reduction in average burst duration is associated with increases in average movement velocity during repetitive flexion and extension of the wrist when modulating the stimulation intensity. Expanding on their approach, we here investigated how subthalamic stimulation with an optimal intensity affects the specific relationship between single beta bursts and their immediate effects on the velocity dynamics, as presented during free drawing. In this context, DBS increased the acceleration around beta bursts,

modifying their immediate ramifications on velocity dynamics during drawing in PD. The modulation of such relationship under stimulation corroborates the notion that DBS-related modulations of beta bursts might underlie the improvement of the progressive deterioration of velocity during ongoing movement in chronically implanted patients.⁴⁷ However, the acute intake of levodopa also reduces duration, amplitude, and rate of subthalamic beta bursts,⁴⁶ and it remains to be elucidated how the effect of DBS might be modulated by levodopa.

Finally, the differential effects of bursts in relationship to the two tasks of drawing freely and with the help of a template highlights the necessity to thoroughly investigate how DBS affects movement under different conditions, and emphasizes the importance of developing new systems and algorithms that can enhance motor function even more effectively. This point is further elaborated in our response to comment 6.

1. It was unclear to me, why the authors decided to include the “template-based” condition, as they did not analyze this condition further with respect to accuracy or error-rates – although this is addressed in the discussion as rationale behind including this condition. I would either considering removing the template condition or include additional analyses with respect to neurophysiological read-outs of behavioral metrics such as accuracy.

Response:

We thank the reviewer for the suggestion to explain more in depth why we included the template-guided drawing condition. We improved this in the revised version of the manuscript (see below) and conducted additional analyses with respect to neurophysiological read-outs of drawing accuracy. Expanding on previous discoveries focused on stereotyped repetitive movements, we here aimed to investigate activities that impose greater demands on fine motor control. To adjust the level of required accuracy, we further incorporated two conditions that were designed to pose different demands on motor control. As such, the two conditions also exhibited variations in how visual feedback informs and directs the planning and execution of movements (see also previous reply and revisions in the introduction on page 4).

Prompted by the reviewer’s comment we have now analyzed two additional parameters of spatial accuracy. We performed radius-angle-transformations of the drawn spirals (San Luciano et al., 2016), calculated linear fits, and assessed their root mean square error (RMSE) and slope as parameters of deviation from an optimal spiral trace and size, respectively. When performing analyses in regards to these parameters, we decided to first include all trials and then test the robustness of our results by excluding those trials in which the pen crossed its own trace as inaccurate (in addition to excluding tremor dominant patients, see comment 4). We added a more detailed description of these analyses in the methods section:

Revisions in the methods section (page 22):

To assess spatial accuracy of motor performance, we first 'unraveled' the original drawings, converting them from the X-Y-representation to the radius-angle-transformation (Supplementary Figure 1).⁵⁵ Next, we fitted a linear model to each radius-angle-transformation and calculated the root mean squared error (RMSE) and the slope of the regression line. Because the increase of radius is constant for Archimedean spirals after the first revolution, we excluded the first revolution (-0.5 π rad to 1.5 π rad) from the linear model. The RMSE represents the average deviation from an optimal trace, while the slope represents the average increase of the radius.

Revisions in the methods section (page 25):

We applied separate linear mixed-effect models using lme4⁶⁸ to investigate the fixed effects of drawing condition (*free vs template*), stimulation (*off vs on*), as well as their interaction effects on the average drawing velocity, RMSE, and slope of radius-angle-transform as dependent variables. To account for within-subject variations each subject had its own random intercept.

Linear mixed effect models with stimulation and drawing condition as independent factors showed that both RMSE and slope were significantly different between the two drawing conditions ($P = 0.001$), supporting the hypothesis that template-guided drawing poses increased accuracy constraints. Stimulation affected the slope ($P = 0.002$), but not the RMSE ($P = 0.051$). We added these results in the paragraph describing the behavioral effects of DBS and drawing condition:

Revisions in the results section (page 7):

Next, we tested if providing a template and applying stimulation affected spatial features of drawing. To this end, we calculated the radius-angle-transformation (Supplementary Fig. 1), fitted linear models for these transformations, and calculated their root-mean-square errors (RMSE) and slopes as parameters of deviation from an optimal spiral. The RMSE was increased when drawing freely ($P < 0.001$), but not affected by stimulation ($P = 0.051$). There was no evidence in favor of an interaction effect ($P = 0.059$). The main effect of drawing condition remained significant when excluding inaccurate trials and when excluding tremor patients (both $P < 0.001$).

The slope of the fitted model was reduced when drawing freely ($P < 0.001$), indicating that patients drew larger spirals when guided by a template. Furthermore, stimulation increased the slope ($P = 0.002$). Both effects remained significant when excluding inaccurate trials (both $P < 0.001$) as well as tremor-dominant patients (effect of drawing condition: $P < 0.001$; effect of stimulation: $P = 0.031$).

We also show that RMSE and the slope directly affect the velocity. The effects of stimulation and drawing condition on velocity remained mostly significant when

accounting for these factors. Only when excluding tremor-dominant patients and simultaneously controlling for the RMSE, the effect of stimulation was not significant anymore ($P = 0.055$). Thus, we added the following paragraph to the results section:

Revisions in the results section (page 7):

To test if spatial parameters influenced the velocity, we separately added the RMSE and the slope as fixed factors to our original model. While both RMSE and the slope were positively associated with the velocity (both $P < 0.001$), the main effects of drawing and stimulation condition remained significant (effect of drawing: both $P < 0.001$ when controlling for RMSE and slope, respectively; effect of stimulation: both $P = 0.04$ when controlling for RMSE and slope, respectively). There were no interactions between stimulation and drawing condition. These results were not affected by excluding inaccurate trials (effect of drawing: both $P < 0.001$ when controlling for RMSE and slope, respectively; effect of stimulation: $P = 0.003$ and $P = 0.039$ when controlling for RMSE and slope, respectively). However, the effect of stimulation did not remain significant when excluding tremor-dominant patients and controlling for RMSE ($P = 0.055$). The effects of drawing condition were not affected when excluding tremor-dominant patients (both $P < 0.001$ when controlling for RMSE and slope, respectively).

These results show that patients generally drew slower when a template was provided, which could be attributed to an increase of accuracy constraints. Applying DBS in the STN sped up drawing movements.

We added a brief paragraph in the discussion that summarizes that our findings support the presence of a speed-accuracy trade-off during drawing:

Revisions in the discussion section (page 15):

Supporting the presence of such a speed-accuracy trade-off, our results demonstrate that the presentation of the template was accompanied by reductions in deviations from an optimal trace and increases in the spiral's diameter. Importantly, RMSE and slope were directly associated with the drawing velocity.

We also examined how excluding inaccurate trials influenced our analyses of beta bursts, which occasionally led to slight adjustments in our findings. As an example, the interaction effect between movement interval and stimulation for burst duration was not significant after excluding inaccurate trials (and also tremor-dominant patients, see comment 4). However, since the relevant pairwise comparisons in the model containing all trials (rest_off – rest_on, draw_off – draw_on) were not significant, this did not influence the further analysis and interpretation. This was amended in the revised version of the manuscript.

Revisions in the results section regarding burst duration (page 9):

We additionally conducted post hoc tests analogously to the analyses of the drawing parameters. Excluding inaccurate trials did not alter the main effect of

drawing interval ($P < 0.001$) but the interaction effect ($P = 0.130$). Excluding seven tremor-dominant patients did not alter the main effect of drawing interval ($P < 0.001$), but the interaction effect ($P = 0.098$).

Revisions in the results section regarding burst amplitude (page 10):

We additionally conducted post hoc tests analogously to the analyses of the drawing parameters. When excluding inaccurate trials, the main effects of movement interval ($P < 0.001$) and stimulation ($P < 0.001$) remained significant. However, there was no interaction between stimulation and the drawing interval ($P = 0.052$). Excluding seven tremor-dominant patients did not alter the main effects of movement interval ($P < 0.001$), stimulation condition ($P < 0.001$), nor the interaction between stimulation and movement interval ($P = 0.034$).

Revisions in the results section regarding the number of bursts (page 10):

We additionally conducted post hoc tests analogously to the analyses of the drawing parameters. When excluding inaccurate trials or tremor-dominant patients, the main effects of movement interval (both $P < 0.001$) and the interaction between movement interval and drawing condition (both $P < 0.001$) remained significant.

Revisions in the results section regarding the number of bursts (page 10):

We additionally conducted post hoc tests analogously to the analyses of the drawing parameters. When excluding inaccurate trials or tremor-dominant patients, the main effect of movement interval remained significant ($P < 0.001$ and $P = 0.016$, respectively).

Regarding the relationship between single beta bursts and immediate reductions in acceleration in the main analysis, excluding inaccurate trials affected the interaction between time interval and drawing condition ($P = 0.057$):

Revisions in the results section (page 11):

The results were not different when excluding tremor-dominant patients (both main effects and both interaction effects remained at $P < 0.05$). Apart from the interaction between time interval * drawing condition ($P = 0.057$), the results were similar when excluding inaccurate trials (main effects of stimulation and drawing condition and interaction between stimulation condition * drawing condition: all $P < 0.05$).

Finally, we conducted an analysis similar to the one that assessed the immediate impact of beta bursts on velocity dynamics to investigate if beta bursts are relevant for spatial deviations from an optimal trace. Instead of the velocity signal, we here extracted the residuals of the fitted models of each spirals' radius-angle

transformation. We report these tests in the paragraph "Beta bursts affect velocity dynamics during drawing":

Revisions in the results section (page 11):

To investigate if the occurrence of a burst not only accompanies the velocity dynamics but also deviations from the optimal spiral trace, we extracted the residuals of the fitted models of each spirals' radius-angle transformation. Similarly to the evaluation of the velocity dynamics, we tested if the instantaneous deviation from the optimal trace differed between the two intervals, and if it was affected by the stimulation or drawing conditions. We did not find any main effects or any interaction effects (all $P > 0.05$), showing that the occurrence of a burst does not directly affect deviations from an optimal drawing trace. The results were not different when controlling for bursting activities in the baseline interval.

Together, the results verify the presence of a speed accuracy trade-off in the two performed tasks. However, we could not find any evidence corroborating a direct relationship between single beta bursts and immediate deviations from an optimal spiral trace.

2. I would welcome it, if the authors would show the absolute durations of beta bursts per condition. Given that they defined beta bursts as 75th percentile above the amplitude distribution of the rest period within the same condition, it is likely that the detected bursts during movement in the ON-DBS condition are much shorter than in the OFF-DBS condition – which could explain the lack of a significant link between deceleration and beta bursting in this condition.

Response:

We appreciate the reviewer's concern and want to point to our supplementary Figures 3-5 (see also below) that were added to plot data without the log transformation and now additionally present the means and standard deviations (free drawing without stimulation: draw: $0.246\text{ s} \pm 0.068$ (mean \pm standard deviation), rest: $0.323\text{ s} \pm 0.220$; free drawing with stimulation: draw: $0.264\text{ s} \pm 0.073$, rest: $0.291\text{ s} \pm 0.087$; template-guided drawing without stimulation: draw: $0.263\text{ s} \pm 0.048$, rest: $0.328\text{ s} \pm 0.176$; template-guided drawing with stimulation: draw: $0.264\text{ s} \pm 0.057$, rest: $0.315\text{ s} \pm 0.238$).

Revised Supplementary Figure 3 (page 46):

Supplementary Figure 3. Burst duration is reduced during drawing in comparison to the rest interval. Single trials, distributions, mean and standard deviation of the mean burst duration are plotted for the different conditions and intervals (blue = free drawing without stimulation, draw: $0.246 \text{ s} \pm 0.068$ (mean \pm standard deviation), rest: $0.323 \text{ s} \pm 0.220$; red = free drawing with stimulation, draw: $0.264 \text{ s} \pm 0.073$, rest: $0.291 \text{ s} \pm 0.087$; green = template-guided drawing without stimulation, draw: $0.263 \text{ s} \pm 0.048$, rest: $0.328 \text{ s} \pm 0.176$; orange = template-guided drawing with stimulation, draw: $0.264 \text{ s} \pm 0.057$, rest: $0.315 \text{ s} \pm 0.238$). We found a significant effect of movement interval ($P < 0.001$).

Prompted by the reviewer's comment we now also directly compared the burst duration within the drawing intervals off and on stimulation and did not find any differences ($P = 1$ for both absolute and log-transformed burst duration). That is, the detected bursts during movement in the ON-DBS condition were not shorter than in the OFF-DBS condition. We now elaborate the results about beta burst duration in more detail in the discussion section:

Revisions in the discussion section (page 16):

The modulatory effect of continuous subthalamic stimulation on burst duration is less clear. In line with our results, Tinkhauser et al.²⁴ showed that conventional DBS does not alter burst duration during rest. Using a different approach for burst detection and a more rigorous withdrawal from medication, Kehnemouyi et al.³² showed that DBS reduces burst duration during repetitive flexion-extension movements of the wrist in chronically stimulated PD patients. Thus, it could be

possible that residual effects of medication or a surgery-related temporal lesioning effect³⁵ masks the modulation of beta burst duration.

Revised Supplementary Figure 4 (page 47):

Supplementary Figure 4. Burst amplitude is reduced during drawing in comparison to the rest interval and reduced by deep brain stimulation. Single trials, distributions, mean and standard deviation of the mean burst amplitude are plotted for the different conditions and intervals (blue = free drawing without stimulation, draw: $0.952 \text{ au} \pm 0.686$ (mean \pm standard deviation), rest: $1.266 \text{ au} \pm 1.134$; red = free drawing with stimulation, draw: $0.874 \text{ au} \pm 0.371$, rest: $0.916 \text{ au} \pm 0.401$; green = template-guided drawing without stimulation, draw: $0.944 \text{ au} \pm 0.690$, rest: $1.204 \text{ au} \pm 1.036$; orange = template-guided drawing with stimulation, draw: $0.818 \text{ au} \pm 0.348$, rest: $0.909 \text{ au} \pm 0.474$). We found significant effects of movement interval ($P < 0.001$), stimulation ($P < 0.001$), and a significant interaction between stimulation and movement interval ($P = 0.020$). The top lines indicate the results from the significant post hoc tests, combined across the drawing conditions (free and template): a) rest_off > draw_off, b) rest_off > rest_on, c) rest_off > draw_on, d) rest_on < draw_off, e) draw_off > draw_on. * $P < 0.05$ (Bonferroni corrected).

Revised Supplementary Figure 5 (page 48):

Supplementary Figure 5. Number of bursts. Single trials, distributions, mean, and standard deviation of the number of bursts are plotted for the different conditions and intervals (blue = free drawing without stimulation, draw: 5.819 ± 3.373 (mean \pm standard deviation), rest: 4.354 ± 1.697 ; red = free drawing with stimulation, draw: 6.627 ± 3.505 , rest: 4.474 ± 1.559 ; green = template-guided drawing without stimulation, draw: 11.720 ± 6.973 , rest: 4.326 ± 1.692 ; orange = template-guided drawing with stimulation, draw: 11.389 ± 6.615 , rest: 4.474 ± 1.655). We found a significant effect of movement interval ($P = 0.001$) as well as an interaction between drawing condition and movement interval ($P < 0.001$). The top lines indicate the results from the significant post hoc tests, combined across the stimulation conditions (on and off): a) free_rest < free_draw, b) free_rest < template_draw, c) free_draw > template_rest, d) free_draw < template_draw, e) template_rest < template_draw. * $P < 0.05$ (Bonferroni corrected).

3. In the discussion, the authors mention that previous bursting could explain movement slowness. If they think that, they should include this feature in their analysis.

Response:

Following the reviewer's suggestion, we added number of bursts, burst rate, burst amplitude, and burst duration from the baseline intervals in our analysis on the

relationship between beta bursts and velocity dynamics as fixed effects. None of the parameters were significantly related to the acceleration, and the main effects and interactions of the original models were not affected by their inclusion. We have added the following paragraph in the results section:

Revisions in the results section (page 11):

To account for the potential influence of preceding bursting activities on the velocity of movement,²⁶ we additionally tested if including the burst duration, burst amplitude, and burst rate of the baseline interval as fixed factors changes the results, which was not the case (both main effects and both interaction effects remained at $P < 0.05$). Furthermore, neither of these parameters had an effect on the slope of the velocity (all $P > 0.05$).

We also tested the effects of previous bursting on the new analysis about the relationship between beta bursts and deviations from the optimal spiral trace, again showing no significant effects:

Revisions in the results section (page 11):

To investigate if the occurrence of a burst not only accompanies the velocity dynamics but also deviations from the optimal spiral trace, we extracted the residuals of the fitted models of each spirals' radius-angle transformation. Similarly to the evaluation of the velocity dynamics, we tested if the instantaneous deviation from the optimal trace differed between the two intervals, and if it was affected by the stimulation or drawing conditions. We did not find any main effects or any interaction effects (all $P > 0.05$), showing that the occurrence of a burst does not directly affect deviations from an optimal drawing trace. The results were not different when controlling for bursting activities in the baseline interval (all $P > 0.05$).

Finally, we also tested whether previous bursting would affect the analysis on the associations between burst amplitude/duration and the extent of changes in acceleration. Here, we found that beta burst duration in the interval before drawing was significantly associated with the extent of acceleration reduction following the occurrence of a burst:

Revisions in the results section (page 12):

Lastly, the results remained similar when controlling for bursting activities in the baseline interval. Interestingly, the average burst duration before drawing was generally associated with the extent of acceleration reductions following a single burst ($P = 0.045$).

In the second linear mixed effect model testing if beta burst duration of bursts occurring during drawing is associated with the extent of acceleration alterations, we did not find any significant main effects of burst duration or any interactions between stimulation, drawing condition, and burst duration (all $P > 0.05$). This was not affected by adding preceding burst activities as fixed effects in the model.

Together, these results demonstrate that beta bursts parallel reductions in acceleration when drawing freely. Additionally, the implementation of DBS specifically modifies movement acceleration around beta bursts in this drawing context.

We further added the following to the discussion section:

Revisions in the discussion section (page 14-15):

Although these results suggest that the occurrence of beta bursts immediately relates to the velocity dynamics of free drawing movements, we did not find a link between the extent of decreased acceleration and neither the burst amplitude nor burst duration. In contrast, we found that the average beta burst duration in the rest interval before drawing generally affected the extent of acceleration changes.

Revisions in the discussion section (page 18):

However, in the current study we did not find that bursts occurring before the drawing task influenced the general acceleration during drawing.

4. Did any of the patients have tremor during drawing? In 7 patients, the main symptom was indicated as being tremor, thus, the authors should investigate how tremor-dominant patient affected their results – and whether they remain robust after removing tremor patients or how performance with DBS changed in these patients.

Response:

Many thanks for raising this important point. Mainly we did not observe effects of tremor during drawing. Only one patient (Patient 3) occasionally presented a slight tremor during drawing (see examples in Figure below).

Examples of spiral-drawings. Four template-guided drawings (each two trials with stimulation turned off and on) of a patient without action tremor are shown on the left, while four template-guided drawings (each two trials with stimulation turned off and on) of patient 3 who experienced occasional action tremor are shown on the right.

Further as recommended, we repeated the main-analyses excluding the mentioned seven tremor-dominant patients. These results are presented in the revised paragraphs that include our supplementary analyses for assessing the robustness of our findings (see also comment 1). Excluding these patients did not affect our behavioral results except the effect of stimulation on velocity when additionally adding RMSE in the models.

Regarding the burst analysis, excluding tremor patients did not affect our results on burst amplitude, burst rate, or number of bursts. However, the interaction effect between movement interval and stimulation for burst duration did not remain significant after excluding seven tremor-dominant patients. Since the relevant pairwise comparisons in the model containing all trials (rest_off – rest_on, draw_off – draw_on) were not significant, this did not influence the further analysis.

Regarding the association between beta bursts and velocity dynamics, excluding tremor dominant patients did not affect the results:

The revisions regarding these analyses in the results section (page 7, 9 and 11) are shown in our reply to comment 1.

*To summarize, our results are robust when excluding the seven tremor-dominant patients. Only when including the RMSE in the mixed effects models testing the effects of stimulation and drawing condition on drawing velocity and simultaneously excluding the seven tremor-dominant patients, the effect of stimulation was not significant anymore ($P = 0.055$). The two-way interactions between stimulation * drawing condition and time interval * drawing condition regarding the associations between beta bursts and velocity dynamics remained significant, highlighting that excluding tremor-dominant patients does not affect our main findings regarding immediate effects of beta bursts on the velocity dynamics.*

5. In the same regard, I noticed that 3 patients did not have any DBS effect and 1 patient a very minor DBS-effect. This is a relevant number in this (relatively) small cohort, and the authors should at least show that these patients still had a task-specific behavioral DBS-effect that may not have been captured in the clinical domain (which might be linked to the very short wash-in period of 2 minutes stated by the authors).

Response:

Following the reviewer's suggestion, we looked into the three patients without DBS effects in more detail. We further identified two patients with minor DBS effects (defined as stim on - stim off ≤ 3), which we also investigated in more detail. Two of these five patients still had a subclinical effect on drawing performance. We report our findings in the results section, Supplementary Table 2 and in Supplementary Figure 6:

Revisions in the results section (page 13):

Five participants showed only minor or no clinical improvement of total UPDRS_{III} when receiving DBS during the study. This might be related to the brief time interval between study participation and the DBS surgery, where micro-lesions might improve motor function even in the absence of stimulation.³⁵ Such lesions might have induced a temporary ceiling effect, where stimulation does not further improve motor function. Another explanation could be the relatively short wash-in and wash-out periods in between conditions. Finally, the electrode position might not have been optimal in these patients.³⁶ In two of these five patients, subclinical effects congruent with our behavioral findings could still be detected when being stimulated. In two patients, drawing performance was not affected and in one patient drawing velocity was reduced. The detailed results are presented in Supplementary Table 3 and the electrode locations modeled by Lead-DBS (Version 2.5.2, <https://www.lead-dbs.org/>)³⁷ are presented in Supplementary Figure 6.

Supplementary Figure 6. Five participants (patients 5, 6, 10, 14, and 18) showed only minor or no clinical improvement of total UPDRS_{III} when receiving DBS during the study. To test if stimulation yielded subclinical effects on drawing execution, we performed linear models with velocity and RMSE as dependent variables and stimulation and drawing conditions as independent variables for each of these subjects individually. Congruent with our behavioral findings when analyzing the whole cohort, patients 14 and 18 showed increased drawing velocity ($P < 0.001$ and $P = 0.002$, respectively), an increased RMSE ($P < 0.001$ and $P = 0.003$, respectively), and an increased slope ($P < 0.001$ and $P = 0.019$, respectively) when being stimulated. We did not find any effects of stimulation on velocity or RMSE in patients 6 and 10 ($P > 0.1$). DBS slowed down drawing speed in patient 5 ($P < 0.001$).

while decreasing RMSE ($P < 0.001$). In this patient, the electrode was positioned slightly outside the STN.

6. The authors state that their results will be helpful for clinical deep brain stimulation, although they show that DBS annuls the link between deceleration and beta bursts. How should this finding then be helpful?

Response:

We thank the reviewer for pointing out that this interpretation needs a more detailed elaboration. Our results demonstrate that distinct effects of DBS on the relationship between single beta bursts and the velocity dynamics depend on the specific context in which DBS is applied. While conventional DBS modulates the relationship between beta bursting and reductions in acceleration in one context, this relationship may not be present in another context (for example by providing a template, as shown here). Because the role of beta bursts may thus differ for movement under different conditions, the observed DBS-related increase in drawing velocity during template-guided drawing is not mediated by modulating the relationship between beta bursts and acceleration. Accordingly, identifying and targeting additional relevant mechanisms for different movement conditions should be helpful in extending and optimizing clinical deep brain stimulation. We have included this interpretation in the discussion section:

Revisions in the discussion section (page 18):

Together, the differential effects of bursts in relationship to our two tasks highlight the necessity to thoroughly investigate how different types of movement can be impacted by DBS. Our results suggest that bursts occurring during free drawing are paralleled by reductions in acceleration, a relationship that does not seem to be present during template-guided drawing. Furthermore, continuous DBS generally increases the acceleration around beta bursts during free drawing, thus ameliorating the impact of subthalamic beta bursts on the velocity dynamics. Such context-dependent modulation of burst activities needs to be considered in the development of future stimulation systems. For example, our results also highlight that stimulation generally increased the drawing velocity in both conditions, which could not be explained by the direct associations between beta bursts and velocity dynamics. Thus, additional factors are involved in determining movement speed,⁵² and it is conceivable that the context in which movement occurs determines the relevant mechanisms that can be targeted to optimally improve motor function.

Response to editorial comments

POLICIES AND FORMS REQUIRED FOR RESUBMISSION

* Nature journals have recently announced an update to our guidance on reporting on sex and gender in research studies (see here). We strongly encourage researchers to follow the 'Sex and Gender Equity in Research – SAGER – guidelines' and to include sex and gender considerations for studies involving humans, vertebrate animals and cell lines where relevant to the topic of study (an overview can be found here). Authors should use the terms sex (biological attribute) and gender (shaped by social and cultural circumstances) carefully in order to avoid confusing both terms.

When preparing your revised manuscript, please be aware of our guidance on Sex and Gender reporting).

Please note that we require that the following recommendations from the guidelines are followed:

1. If the research findings apply to only one sex or gender, that must be indicated in the title and/or abstract.

Response:

Our findings apply to both sexes and genders.

2a. For studies involving vertebrates animal and cell lines- The Reporting Summary should include whether sex was considered in the study design.

Response:

Our study involves human research participants and we did not specifically consider sex or gender (see also point 2b)

2b. For studies involving human research participants- The Reporting Summary should include whether sex and/or gender was considered in the study design and whether sex and/or gender of participants was determined based on self-report or assigned (and methodology used).

Response:

Sex and gender were not specifically considered in the study design. Sex was assigned to the patients. Please also see the reporting summary:

Participants of both biological sexes were included (15 male and 4 female patients). No consent has been obtained for reporting individual level data. Because the aim of the study is to investigate the role of basal ganglia beta

oscillations for fine controlled motor skills in Parkinson's disease and because the sample size is too low to quantify potential sex differences we did not conduct additional sex- and gender-based analyses. In the source data file, the data are shown excluding information on sex or gender.

3. Data should be reported disaggregated for sex and gender where this information has been collected and consent has been obtained for reporting and sharing individual-level data; disaggregated numbers for individual experiments must be provided in the source data as appropriate whereas overall numbers may be provided in the Nature Portfolio Reporting Summary.

Response:

Overall numbers about sex and gender are now provided in Table 1 and in the reporting summary.

Information on the points above should be included in the revised manuscript and detailed in the cover letter.

In addition, please note that if sex- and gender-based analyses have been performed a priori, results should be reported regardless of positive or negative outcome. We discourage conducting post hoc sex- and gender-based analysis if the study design is insufficient (for example, low sample size) to enable meaningful conclusions.

If no sex- and gender-based analyses have been performed, please indicate the reasons for the lack of these analyses in the Reporting Summary.

Response:

We indicate the reason for not performing any sex- and gender-based analysis in the reporting summary:

Because the aim of the study is to investigate the role of basal ganglia beta oscillations for fine controlled motor skills in Parkinson's disease and because the sample size is too low to quantify potential sex differences we did not conduct additional sex- and gender-based analyses.

* Your paper uses custom code/software. Please complete the following code and software submission checklist and make your code available for reviewer assessment, if you have not already done so. The code/software can be provided in a zip file with a readme.txt file or other instructions for installing and running the software. If appropriate, also provide example data and expected output. If you have any issues with the file upload, please let me know.

<https://www.nature.com/documents/nr-software-policy.pdf>

Response:

We have completed the code and software submission checklist and provide our code including exemplary data and a readme.txt in an attached SpiralBeta-main.zip file. The code will also be made available on <https://github.com/manubange/SpiralBeta.git>

DATA AND CODE AVAILABILITY

* All Nature Communications manuscripts must include a "Data Availability" section after the Methods section but before the References. If any of the data can only be shared on request or are subject to restrictions, please specify the reasons and explain how, when, and by whom the data can be accessed. For more information on this policy and a list of examples, see:

<https://www.nature.com/documents/nr-data-availability-statements-data-citations.pdf>

Response:

We have included the "Data Availability" statement after the methods section:

The data that support the findings of this study are available upon reasonable request from the corresponding author. Participant consent does not allow for openly depositing the full original dataset online. Source data are provided with this paper.

* Please also include a "Code Availability" section after the "Data Availability" section. If the code can only be shared on request, please specify the reasons. For more information on our code sharing policy and requirements, please see:

<https://www.nature.com/nature-portfolio/editorial-policies/reporting-standards#availability-of-computer-code>

Response:

We have included the "Code Availability" statement after the methods section:

All relevant codes employed in the study can be freely accessed without restriction at <https://github.com/manubange/SpiralBeta.git>

* As Nature Portfolio policies strongly encourage you to share your research data in a public repository (e.g. spreadsheets, text, images), we are partnering with the figshare repository so that you can use the figshare integration via the 'Research Data Deposition' tab when submitting your revised manuscript.

Data are stored privately until a manuscript decision is reached and you can edit/withdraw them up to this point: you retain rights and control over your data. The data will be published at the same time as your article; you will receive a data DOI, with guidance on linking the data and manuscript. In the event your manuscript is not accepted, you can keep or remove your data in figshare.

We recommend the use of discipline-specific repositories where available and for a number of data types this is mandatory. Ensure you do not submit these data types or any sensitive data to figshare.

Response:

The data that support the findings of this study are available upon reasonable request from the corresponding author. Participant consent does not allow for openly depositing the full original dataset online. Source data are provided with this paper.

* We strongly encourage you to deposit all new data associated with the paper in a persistent repository where they can be freely and enduringly accessed. We recommend submitting the data to discipline-specific and community-recognised repositories; a list of repositories is provided here:

<http://www.nature.com/sdata/policies/repositories>

Refer to our data policies here:

<https://www.nature.com/nature-portfolio/editorial-policies/reporting-standards#availability-of-data>

Response:

The data that support the findings of this study are available upon reasonable request from the corresponding author. Participant consent does not allow for openly depositing the full original dataset online. Source data are provided with this paper.

* To maximise the reproducibility of research data, we strongly encourage you to provide a file containing the raw data underlying the following types of display items:

- Any reported means/averages in box plots, bar charts, and tables
- Dot plots/scatter plots, especially when there are overlapping points
- Line graphs

The data should be provided in a single Excel file with data for each figure/table in a separate sheet, or in multiple labelled files within a zipped folder. Name this file or folder 'Source Data', and include a brief description in your cover letter. The

"Data Availability" section should also include the statement "Source data are provided with this paper."

To learn more about our motivation behind this policy, please see:

<https://www.nature.com/articles/s41467-018-06012-8>

Response:

We uploaded the source data for all figures and supplementary figures and added the statement "Source data are provided with this paper." in the "Data Availability" section. The File contains the raw data underlying Figures 2a, 2c, 3,4,6a, and Supplementary Figures 3,4, and 5 in separate sheets.

* Please replace your bar graphs with plots that feature information about the distribution of the underlying data. All data points should be shown for plots with a sample size less than 10. For larger sample sizes, please consider box-and-whisker or violin plots as alternatives. Measures of centrality, dispersion and/or error bars should be plotted and described in the figure legend.

Response:

We present our data in violin plots including the means, standard deviations, and individual data points.

ORCID

* Nature Communications is committed to improving transparency in authorship. As part of our efforts in this direction, we are now requesting that all authors identified as 'corresponding author' create and link their Open Researcher and Contributor Identifier (ORCID) with their account on the Manuscript Tracking System prior to acceptance. ORCID helps the scientific community achieve unambiguous attribution of all scholarly contributions.

You can create and link your ORCID from the home page of the Manuscript Tracking System by clicking on 'Modify my Springer Nature account' and following these instructions. Please also inform all co-authors that they can add their ORCIDs to their accounts and that they must do so prior to acceptance.

For more information please visit

<http://www.springernature.com/orcid>

If you experience problems in linking your ORCID, please contact the Platform Support Helpdesk.

Response:

The corresponding author's ORCID is <https://orcid.org/0000-0002-2551-5655>

References

- Danna, J., S. Athènes and P. G. Zanone (2011). "Coordination dynamics of elliptic shape drawing: effects of orientation and eccentricity." *Hum Mov Sci* **30**(4): 698-710.
- Danna, J., J.-L. Velay, A. Eusebio, L. Véron-Delor, T. Witjas, J.-P. Azulay and S. Pinto (2018). "Digitalized spiral drawing in Parkinson's disease: A tool for evaluating beyond the written trace." *Human Movement Science*.
- Kehnemouyi, Y. M., M. N. Petrucci, K. B. Wilkins, J. A. Melbourne and H. M. Bronte-Stewart (2023). "The Sequence Effect Worsens Over Time in Parkinson's Disease and Responds to Open and Closed-Loop Subthalamic Nucleus Deep Brain Stimulation." *J Parkinsons Dis* **13**(4): 537-548.
- Kehnemouyi, Y. M., K. B. Wilkins, C. M. Anidi, R. W. Anderson, M. F. Afzal and H. M. Bronte-Stewart (2021). "Modulation of beta bursts in subthalamic sensorimotor circuits predicts improvement in bradykinesia." *Brain* **144**(2): 473-486.
- Lofredi, R., W.-J. Neumann, A. Bock, A. Horn, J. Huebl, S. Siegert, G.-H. Schneider, J. K. Krauss and A. A. Kühn (2018). "Dopamine-dependent scaling of subthalamic gamma bursts with movement velocity in patients with Parkinson's disease." *eLife* **7**: e31895.
- Lofredi, R., H. Tan, W. J. Neumann, C. H. Yeh, G. H. Schneider, A. A. Kuhn and P. Brown (2019). "Beta bursts during continuous movements accompany the velocity decrement in Parkinson's disease patients." *Neurobiol Dis* **127**: 462-471.
- San Luciano, M., C. Wang, R. A. Ortega, Q. Yu, S. Boschung, J. Soto-Valencia, S. B. Bressman, R. B. Lipton, S. Pullman and R. Saunders-Pullman (2016). "Digitized Spiral Drawing: A Possible Biomarker for Early Parkinson's Disease." *PLoS One* **11**(10): e0162799.
- Torrecillos, F., S. He, A. A. Kühn and H. Tan (2023). "Average power and burst analysis revealed complementary information on drug-related changes of motor performance in Parkinson's disease." *NPJ Parkinsons Dis* **9**(1): 93.
- Torrecillos, F., G. Tinkhauser, P. Fischer, A. L. Green, T. Z. Aziz, T. Foltynie, P. Limousin, L. Zrinzo, K. Ashkan, P. Brown and H. Tan (2018). "Modulation of Beta Bursts in the Subthalamic Nucleus Predicts Motor Performance." *The Journal of neuroscience : the official journal of the Society for Neuroscience* **38**(41): 8905-8917.

REVIEWERS' COMMENTS

Reviewer #1 (Remarks to the Author):

They have addressed all of my comments and I really like this paper a lot. No concerns.

Reviewer #2 (Remarks to the Author):

The Authors addressed correctly my comments. I have no further comments.

Reviewer #3 (Remarks to the Author):

The authors corrected the manuscript, which is now excellent for publishing in the journal.

Reviewer #4 (Remarks to the Author):

The authors addressed all my previously risen points extensively and I have no further requests regarding a revised version of the manuscript.

Response to the reviewers' comments

We thank the reviewers for their support and constructive feedback to the previous version of our manuscript, which we believe has significantly improved the quality of the paper.

Reviewer #1 (Remarks to the Author):

They have addressed all of my comments and I really like this paper a lot. No concerns.

Response:

We thank the reviewer for the appreciation and the constructive suggestions to the previous version, which we believe have greatly improved the paper.

Reviewer #2 (Remarks to the Author):

The Authors addressed correctly my comments. I have no further comments.

Response:

We express our gratitude to the reviewer for the positive evaluation and the valuable comments on the earlier version of the manuscript, which we are confident have significantly enhanced the quality of the manuscript.

Reviewer #3 (Remarks to the Author):

The authors corrected the manuscript, which is now excellent for publishing in the journal.

Response:

We thank the reviewer for their positive feedback and the helpful comments to the previous manuscript version. We firmly believe that these inputs have significantly enhanced the quality of the paper.

Reviewer #4 (Remarks to the Author):

The authors addressed all my previously risen points extensively and I have no further requests regarding a revised version of the manuscript.

Response:

We thank the reviewer for the positive evaluation and the thoughtful comments to the previous manuscript version, which have significantly improved the paper.